# A bipartite, low-affinity roadblock domain-containing GAP complex regulates bacterial front-rear polarity

**Dobromir Szadkowski, Luís António Menezes Carreira⬀, Lotte Søgaard-Andersen⬀***

Department of Ecophysiology, Max Planck Institute for Terrestrial Microbiology, Marburg, Germany

* sogaard@mpi-marburg.mpg.de

**Data Availability Statement:** All relevant data are within the manuscript and its Supporting Information files.

## Abstract

The Ras-like GTPase MglA is a key regulator of front-rear polarity in the rod-shaped *Myxococcus xanthus* cells. MglA-GTP localizes to the leading cell pole and stimulates assembly of the two machineries for type IV pili-dependent motility and gliding motility. MglA-GTP localization is spatially constrained by its cognate GEF, the RomR/RomX complex, and GAP, the MglB Roadblock-domain protein. Paradoxically, RomR/RomX and MglB localize similarly with low and high concentrations at the leading and lagging poles, respectively. Yet, GEF activity dominates at the leading and GAP activity at the lagging pole by unknown mechanisms. Here, we identify RomY and show that it stimulates MglB GAP activity. The MglB/RomY interaction is low affinity, restricting formation of the bipartite MglB/RomY GAP complex almost exclusively to the lagging pole with the high MglB concentration. Our data support a model wherein RomY, by forming a low-affinity complex with MglB, ensures that the high MglB/RomY GAP activity is confined to the lagging pole where it dominates and outcompetes the GEF activity of the RomR/RomX complex. Thereby, MglA-GTP localization is constrained to the leading pole establishing front-rear polarity.

## Author summary

Bacterial cells are spatially highly organized with proteins localizing to distinct subcellular locations. This spatial organization, or cell polarity, is important for many cellular processes including motility. The rod-shaped *M. xanthus* cells move with defined leading and lagging cell poles. This front-rear polarity is brought about by the polarity module, which consists of the small Ras-like GTPase MglA, its GEF (the RomR/RomX complex) and its GAP (MglB). Specifically, MglA-GTP localizes to the leading pole and stimulates assembly of the motility machineries. MglA-GTP localization, in turn, is spatially constrained by its GEF and GAP. Paradoxically, the RomR/RomX GEF and MglB GAP localize similarly with low and high concentrations at the leading and lagging poles, respectively. Yet, GEF activity dominates at the leading and GAP activity at the lagging pole. Here, we identify RomY and show that it stimulates MglB GAP activity. Interestingly, the MglB/RomY interaction is low affinity. Consequently, MglB/RomY complex formation almost exclusively occurs at the lagging cell pole with the high MglB concentration. Thus, the key to

**Funding:** This work was funded by the Deutsche Forschungsgemeinschaft (project no. 269423233) within the framework of the Transregio 174 'Spatiotemporal dynamics of bacterial cells' (to L. S.-A.) (https://www.dfg.de/) as well as by the Max Planck Society (to L.S.-A.) (https://www.mpg.de/en). The funders had no role in study design, data collection and analysis, decision to publish, or preparation of the manuscript.

**Competing interests:** The authors have declared that no competing interests exist.

precisely stimulating MglB GAP activity only at the lagging pole is that the MglB/RomY interaction is low-affinity, ultimately restricting MglA-GTP to the leading pole.

## Introduction

Cell polarity enables essential cellular processes such as growth, division, differentiation, and motility [1–3]. Small GTPases of the Ras superfamily are key cell polarity regulators in eukaryotes and bacteria [4–8], while they remain underexplored in archaea despite being abundant in several lineages [9]. Typically, the function of small GTPases in cell polarity is coupled to their subcellular localization [4–7]. A central unresolved question is how the precise subcellular localization of these GTPases is established.

Ras superfamily GTPases are molecular switches that alternate between an inactive, GDP-bound and an active, GTP-bound conformation [10]. The nucleotide-dependent conformational changes center on the switch-1 and switch-2 regions close to the nucleotide-binding pocket, allowing the GTP-bound GTPase to interact with downstream effectors to implement a specific response [10]. The activation/deactivation cycle is regulated by a cognate guanine-nucleotide exchange factor (GEF), which facilitates the exchange of GDP for GTP, and a GTPase activating protein (GAP), which stimulates the low intrinsic GTPase activity [11,12]. Generally, the subcellular localization of a small GTPase is brought about by the localized activity of its cognate GEF, while the role played by its cognate GAP is less well-understood [4,8].

Motility in bacterium *Myxococcus xanthus* is an excellent model system to investigate how the spatiotemporal regulation of a small GTPase by its cognate GEF and GAP establishes dynamic cell polarity. *M. xanthus* cells are rod-shaped and translocate across surfaces with defined front-rear polarity, i.e. with a leading and lagging cell pole [7,13]. In response to signaling by the Frz chemosensory system, front-rear polarity is inverted, and cells reverse their direction of movement [14]. Motility and its regulation by the Frz system are prerequisites for multicellular morphogenesis with the formation of spreading, predatory colonies in the presence of nutrients and spore-filled fruiting bodies in the absence of nutrients [7,13]. *M. xanthus* has two polarized motility systems. Gliding motility depends on the Agl/Glt complexes that assemble at the leading pole, adhere to the substratum, and disassemble at the lagging pole [15,16]. In the type IV pili (T4P)-dependent motility system, T4P assemble at the leading pole [17] and undergo extension-adhesion-retraction cycles that pull a cell forward [18,19]. Accordingly, during Frz-induced reversals, the cell pole at which the motility machineries assemble switches [15,17,20].

Front-rear polarity in *M. xanthus* is established by the so-called polarity module that consists of the small cytoplasmic GTPase MglA and its regulators. MglA generates the output of the polarity module and is essential for both motility systems [21,22]. MglA follows the canonical scheme for small GTPases in cell polarity with the active GTP-bound state localizing to the leading cell pole, while the inactive MglA-GDP is diffused in the cytoplasm [23,24]. At the leading pole, MglA-GTP stimulates assembly of the Agl/Glt complexes [16,25,26] and extension of T4P [27,28] by interacting with downstream effectors. The cognate GEF and GAP of MglA control its nucleotide-bound state and localization. The RomR/RomX complex has MglA GEF activity [29]. In this complex, RomX interacts with MglA to stimulate nucleotide exchange, and this activity is enhanced by RomR [29]. Neither RomX nor RomR share homology with known GEFs in eukaryotes [11,12,29]. MglB has MglA GAP activity *in vitro* [23,24].

Structural analyses have demonstrated that MglB is a homodimeric Roadblock domain-containing protein and forms a 2:1 complex with MglA-GTP [30–32].

The RomR/RomX complex and MglB also localize polarly and, unexpectedly, localize in the same bipolar asymmetric pattern with a high concentration at the lagging and a low concentration at the leading pole [23,24,29,33–35]. Nonetheless, *in vivo* evidence supports that GEF activity dominates at the leading pole [29], while GAP activity dominates at the lagging pole [16,27,35,36]. RomR/RomX recruits MglA-GTP to the leading pole via two mechanisms: One depends on GEF activity, and in the second, the RomR/RomX complex interacts directly with MglA-GTP [29]. MglB, via its GAP activity, excludes MglA from the lagging pole [35,36]. Therefore, in the absence of MglB, MglA-GTP localizes more symmetrically at the cell poles [23,24,36], resulting in the formation of T4P at both poles [27] and lack of Agl/Glt complex disassembly at the lagging pole [16]. Consequently, cells lose front-rear polarity, hyper-reverse erratically independently of the Frz system, and display little net movement. During the Frz-induced reversals, the polarity of MglA, MglB and RomR/RomX is inverted [23,24,29,34], thus, laying the foundation for the assembly of the motility machineries at the new leading pole. The mechanism underpinning the spatial separation of the GEF and GAP activities to the two cells poles is unclear.

Here, we investigated how the RomR/RomX GEF and MglB GAP activities are spatially separated. We report the identification of MXAN_5749/MXAN_RS27865 (reannotated from MXAN_5749 to MXAN_RS27865 in the NCBI Reference Sequence NC_008095.1; from hereon RomY) and demonstrate that RomY has MglB GAP stimulating activity. Notably, the MglB/RomY interaction is low affinity, and, therefore, formation of the bipartite MglB/RomY GAP complex occurs almost exclusively at the lagging cell pole with the high MglB concentration. Consequently, MglB GAP activity is stimulated at the lagging pole, thereby restricting MglA-GTP to the leading pole. Thus, the key to precisely stimulating MglB GAP activity at the lagging pole is that the MglB/RomY interaction is low-affinity.

## Results

### RomY is essential for the correct reversal frequency

Using a set of 1611 prokaryotic genomes, we previously used a phylogenomic approach to identify RomX [29]. This approach was based on the observations that MglA and MglB homologs are widespread in prokaryotes [37]. At the same time, RomR has a more narrow distribution and, generally, co-occurs with MglA and MglB [33]. We, therefore, reasoned that proteins with a genomic distribution similar to RomR would be candidates for being components of the polarity module. Using this strategy, we also identified the uncharacterized protein RomY (Fig 1A).

Based on sequence analysis, RomY is a 188-residue cytoplasmic protein. The RomY homologs identified in the 1611 genomes share a conserved N-terminal region, which includes residues 8–89 in RomY of *M. xanthus* and does not match characterized domain models, and a partially conserved C-terminal motif (S1A Fig; S1 Table). The *romY* locus is partially conserved in Myxococcales, but none of the genes flanking *romY* has been implicated in motility (S1B Fig).

To characterize RomY function, we generated a *romY* in-frame deletion mutation (Δ*romY*) in the wild-type (WT) strain DK1622. In population-based motility assays, cells were spotted on 0.5% and 1.5% agar that are favorable to T4P-dependent and gliding motility, respectively [38]. On 0.5% agar, WT displayed long flares at the colony edge characteristic of T4P-dependent motility, while the Δ*pilA* mutant, which cannot assemble T4P, generated smooth colony edges; the Δ*romY* mutant formed shorter flares and had significantly reduced colony

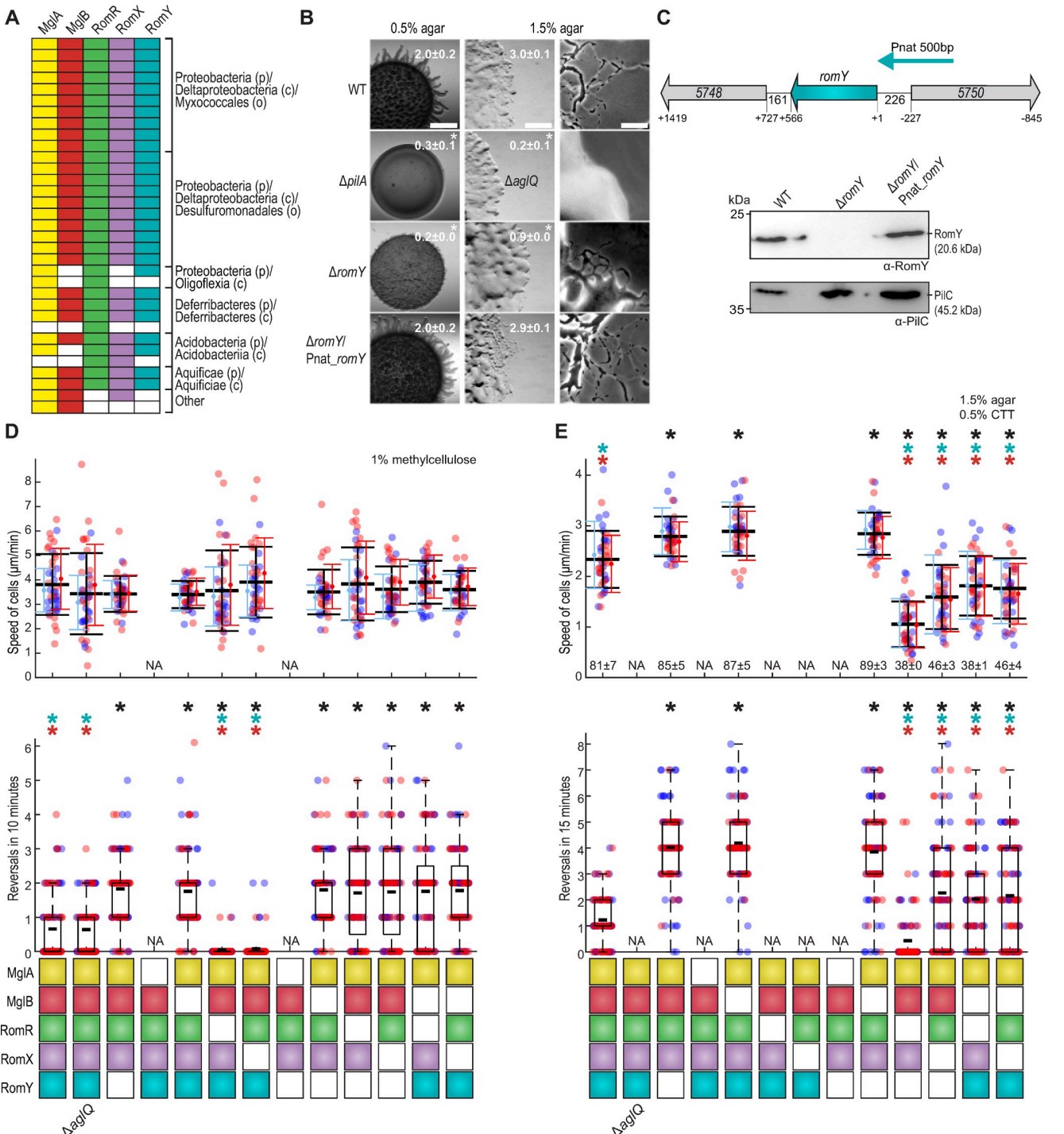

**Fig 1. RomY is a component of the polarity module and important for correct reversals. A.** RomY co-occurs with proteins of polarity module. Each column indicates the presence or absence of the relevant gene for the indicated proteins as colored or white boxes, respectively in a set of 1611 prokaryotic genomes. Lowest taxonomic level that includes all species in a group are indicated as phylum (p), class (c) and order (o). **B.** RomY is important for both motility systems. Cells were incubated on 0.5/1.5% agar with 0.5% CTT to score T4P-dependent/gliding motility. Scale bars, 1mm (left), 500 μm (middle), 50 μm (right). Numbers, colony expansion in mm in 24hrs as mean ± standard deviation (STDEV) ($n = 3$); * $P<0.05$, two-sided Student's $t$-test. **C.** *romY* locus and accumulation of RomY. Upper panel, *romY* locus; numbers in arrows, MXAN locus tags (note that in the NCBI Reference Sequence NC_008095.1, MXAN_5748 and MXAN_5750 are reannotated as MXAN_RS27860 and MXAN_RS27870, respectively); numbers below, distance between stop and start codons. Cyan arrow, 500 bp fragment used for ectopic expression of *romY* and *romY-YFP*. Lower panel, immunoblot analysis of RomY accumulation. Cell lysates prepared from same number of cells were separated by SDS–PAGE and probed with α-RomY antibodies and α-PilC antibodies after stripping (loading

control). **D, E.** RomY is important for correct reversals. Boxes below diagrams indicate the presence or absence of indicated proteins as colored or white boxes, respectively. The Δ*aglQ* mutant is a control that T4P-dependent motility is scored in (**D**) and gliding in (**E**). Individual data points from two independent experiments with each *n* = 20 cells (upper panels) and *n* = 50 cells (lower panels) are plotted in red and blue. Upper diagrams, speed of cells moving by T4P-dependent motility (**D**) or gliding (**E**). Mean±STDEV is shown for each experiment and for both experiments (black). In (**E**), numbers indicate mean fraction ±STDEV of moving cells. NA, not applicable because cells are non-motile. Lower panels, boxplots of reversals per cell in 10 or 15min; boxes enclose 25th and 75th percentiles, thick black line indicates the mean and whiskers the 10th and 90th percentiles. In all panels, * *P*<0.01, two-sided Student's *t*-test. Black, cyan and red * indicate comparison to WT, the Δ*romY* strain and the Δ*mglB* strain, respectively.

expansion compared to WT (Fig 1B). On 1.5% agar, WT displayed single cells at the colony edge characteristic of gliding motility, while the Δ*aglQ* mutant, which lacks a component of the Agl/Glt machinery, did not. The Δ*romY* mutant had fewer single cells at the colony edge and significantly reduced colony expansion compared to WT (Fig 1B). In complementation experiments, ectopic expression of *romY* from its native promoter on a plasmid integrated in a single copy at the Mx8 *attB* site restored the defects in both motility systems (Fig 1B and 1C). Ectopically produced RomY accumulated at a level similar to that in WT (Fig 1C).

Using assays to monitor the motility characteristics with single-cell resolution, we observed that for both motility systems, Δ*romY* cells moved with speeds similar to WT (Fig 1D and 1E, upper panels), but reversed at a significantly higher frequency than WT (Fig 1D and 1E, lower panels). Importantly, in the absence of the FrzE kinase, which is essential for Frz-induced reversals [39], Δ*romY* cells, similarly to Δ*mglB* cells [33,35], still hyper-reversed (S2 Fig).

Altogether, we conclude that RomY is not necessary for motility *per se* but for maintaining the correct reversal frequency. Moreover, the epistasis experiment demonstrating that Δ*romY* cells, similarly to Δ*mglB* cells, hyper-reverse in the absence of the FrzE kinase supports that RomY, similar to MglB, acts downstream of the Frz system to maintain the correct reversal frequency.

## A Δ*romY* mutant has the same phenotype as the Δ*mglB* mutant

We performed epistasis tests using single-cell motility characteristics as readouts to test whether RomY functions in the same genetic pathway as MglA, MglB, RomR and RomX. In T4P-dependent motility (Fig 1D), all single and double mutants except for the Δ*mglA* and the Δ*mglA*Δ*romY* mutants, none of which displayed movement, had speeds similar to WT. The Δ*mglB*, Δ*romY* and Δ*mglB*Δ*romY* mutants had the same hyper-reversing phenotype. As previously reported, the Δ*romR* and Δ*romX* mutants hypo-reversed [29,40], while the Δ*romR*Δ*romY* and Δ*romX*Δ*romY* double mutants had reversal phenotypes similar to that of the Δ*romY* mutant. Because the Δ*mglB* and Δ*romY* single mutants have the same hyper-reversal phenotype and the Δ*mglB*Δ*romY* double mutant did not have an additive hyper-reversal phenotype, we included the previously described [29] Δ*mglB*Δ*romR* and Δ*mglB*Δ*romX* double mutants in our analyses; these two mutants had reversal phenotypes similar to that of the Δ*mglB* mutant and the Δ*romR*Δ*romY* and Δ*romX*Δ*romY* double mutants.

In gliding motility (Fig 1E), the Δ*mglA*, Δ*romR* and Δ*romX* mutants are non-motile because no or insufficient MglA-GTP accumulate to stimulate Agl/Glt complex formation. Again, the Δ*mglB*, Δ*romY* and Δ*mglB*Δ*romY* mutants were similar with respect to speed and also had the same hyper-reversal phenotype. Importantly, the Δ*romY* mutation, similarly to the Δ*mglB* mutation (Fig 1E, [29]), partially alleviated the deleterious effect of the Δ*romR* and Δ*romX* mutations on gliding.

These epistasis experiments support that RomY acts in the same pathway as MglA, MglB, RomR and RomX. Moreover, the epistasis experiments in cells lacking the FrzE kinase support that RomY, similarly to MglB [16,23,24,27,33,35], acts downstream of the Frz system to maintain correct reversals (S2 Fig). Notably, lack of MglB or RomY causes (1) strikingly similar and

non-additive phenotypes (Fig 1D and 1E), (2) Frz-independent hyper-reversals (S2 Fig) and (3) partial suppression of the gliding motility defect in the ΔromR and ΔromX mutants (Fig 1E). The Frz-independent hyper-reversals caused by lack of MglB result from the accumulation of MglA-GTP at both poles with the concomitant loss of front-rear polarity [16,33,35]. The defect in gliding motility in the ΔromR and ΔromX mutants is caused by a lack of MglA GEF activity and, therefore, a low MglA-GTP level. These considerations support that lack of RomY, similarly to lack of MglB, causes increased accumulation of MglA-GTP.

At least three non-mutually exclusive scenarios can explain the increased accumulation of MglA-GTP in the ΔromY mutant: RomY (1) inhibits RomR/RomX GEF activity, (2) stimulates MglB GAP activity, or (3) has MglA GAP activity. These scenarios make different predictions for the epistasis experiments in which the ΔromY mutation is combined with either the ΔmglA, ΔmglB, ΔromR or ΔromX mutation. In scenario (1), the ΔromY mutation would not suppress the gliding defect in the ΔromR and ΔromX mutants, and the effects of the ΔmglB and ΔromY mutations on reversal frequency would be additive. Scenario (2) and (3) predict that the ΔromY mutation would suppress the gliding defect in the ΔromR and ΔromX mutants; however, in scenario (2), the effects of the ΔmglB and ΔromY mutations would not be additive, while they would be additive in scenario (3). The predictions of scenario (1) and (3) are not in agreement with the results of the epistasis experiments (Fig 1D and 1E), while all the results of the epistasis experiments agree with scenario (2), thus, supporting that RomY stimulates MglB GAP activity.

## RomY has MglB GAP stimulating activity *in vitro*

Prompted by the above considerations, we examined the effect of RomY on MglA GTPase activity *in vitro* by measuring either released GDP in a regenerative coupled enzyme assay or released phosphate ($P_i$) in a malachite green-based assay using purified MglA-His$_6$, His$_6$-MglB and Strep-RomY (S3A Fig). 3μM MglA-His$_6$ was preloaded with GTP and then mixed with MglB-His$_6$ and/or Strep-RomY in the presence of ~300-fold molar excess of GTP. MglA-His$_6$ alone had a low GTPase activity of ~2/3 GTP/MglA/hr in the regenerative coupled enzyme/ malachite green-based assays (Fig 2A). 3μM MglA-His$_6$ mixed in a 1:2 ratio with His$_6$-MglB had a ~six-fold increased activity of ~12/18 GTP/MglA/hr in the two assays (Fig 2A) in agreement with MglB having GAP activity. These specific GTPase activities are similar to those previously reported for MglA-His$_6$ alone and MglA-His$_6$:His$_6$-MglB mixed in a 1:2 ratio [29,30,32]. 3μM MglA-His$_6$ mixed in a 1:2 ratio with Strep-RomY did not display increased GTPase activity (Fig 2A). Importantly, 3μM MglA-His$_6$ mixed in a 1:2:2 ratio with MglB-His$_6$ and Strep-RomY displayed an ~two-fold increase in GTPase activity to ~25/31 GTP/MglA/hr in the two assays (Fig 2A). Neither MglB-His$_6$ nor Strep-RomY had GTPase activity. In conclusion, RomY stimulates MglA GTPase activity but only in the presence of MglB.

Next, we investigated how MglA, MglB and RomY interact using pull-down experiments with Strep-RomY as the bait. Strep-RomY alone bound to Strep-Tactin beads, while His$_6$-MglB, MglA-His$_6$-GTP, MglA-His$_6$-GDP and the His$_6$-MalE negative control neither bound alone nor in the presence of Strep-RomY (S3A and S3B Fig). We, therefore, speculated that the interaction(s) between RomY and MglB and/or MglA could be low affinity resulting in transient complex formation. To test this possibility, we added the protein cross-linker dithiobis (succinimidyl propionate) (DSP) to the protein mixtures before affinity chromatography. After elution, samples were separated by SDS-PAGE before and after cross-links were broken with dithiothreitol (DTT).

Cross-linked Strep-RomY alone bound to the Strep-Tactin beads and eluted as a high-molecular weight complex (S3C Fig). Upon DTT treatment of the eluted sample, Strep-RomY

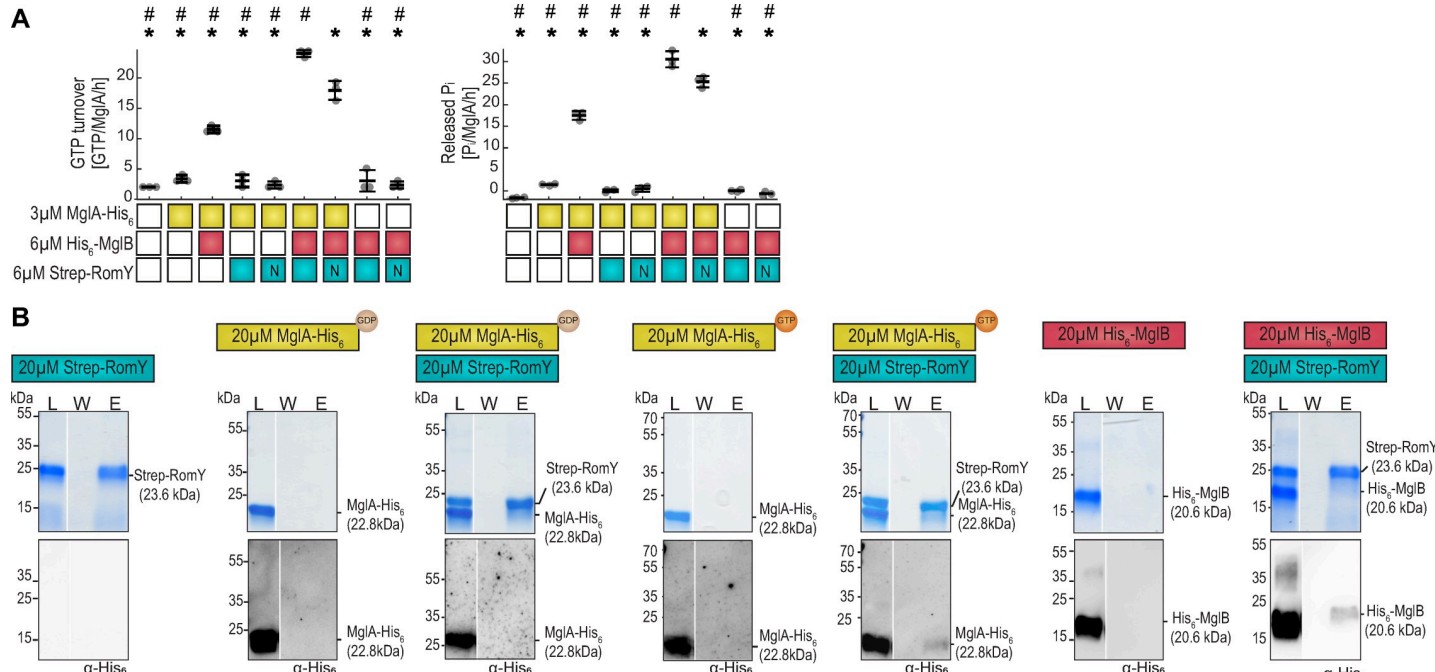

**Fig 2. RomY stimulates MglA GTPase activity in the presence of MglB and interact with MglA-GTP and MglB. A.** RomY stimulates MglA GTPase activity in the presence of MglB. GTPase activity measured as GTP turnover in a regenerative coupled enzyme assay (left) and as released inorganic phosphate in malachite green assay (right), after 1h of incubation. Boxes below diagrams indicate the presence or absence of indicated proteins as colored or white boxes, respectively, GTP was added to 1mM. For Strep-RomY, N indicate Strep-RomY[N]. Individual data points from three independent experiments are in gray and mean±STDEV indicated. * and #, $P<0.05$, two-sided Student's $t$-test with samples compared to MglA-His$_6$-MglB/Strep-RomY and MglA-His$_6$/His$_6$-MglB/Strep-RomY[N], respectively. **B.** RomY interacts with MglB and MglA-GTP. Proteins were mixed with final concentrations and 10mM GTP/GDP as indicated in the schematics for 30min at RT, DSP added (final concentration 200µM, 5min, RT), DSP quenched, and proteins applied to Strep-Tactin coated magnetic beads. Fractions before loading (L), the last wash (W) and after elution (E) were separated by SDS–PAGE, gels stained with Coomassie Brilliant Blue (upper panels) and subsequently probed with α-His$_6$ antibodies (lower panels). All samples were treated with loading buffer containing 100mM DTT to break cross-links before SDS-PAGE. For each combination, fractions were separated on the same gel. Gaps between lanes indicate lanes deleted for presentation purposes. The experiments in S3C Fig are similar to those presented in this panel in which proteins were separated on an 8–16% acrylamide gradient gel; proteins were separated on a 7.5% acrylamide gel in the experiments in S3C Fig to increase the separation of higher molecular weight complexes.

that migrated at the size of a monomer was recovered from the sample (Figs 2B and S3C). In the presence of Strep-RomY and MglA-His$_6$ preloaded with GTP or His$_6$-MglB, high-molecular weight complexes that were recognized by specific α-MglA and α-MglB antibodies were eluted from the Strep-Tactin beads before protein cross-links were broken (S3C Fig). Upon DTT treatment of the eluted samples, MglA-His$_6$ and His$_6$-MglB that migrated at the size of monomers were recovered together with Strep-RomY from these samples (Figs 2B and S3C). By contrast, MglA preloaded with GDP and cross-linked in the presence of Strep-RomY was not retained on the Strep-Tactin beads (Figs 2B and S3C). Finally, neither cross-linked His$_6$-MalE, MglB-His$_6$ nor MglA-His$_6$ preloaded with GTP or GDP alone bound to the Strep-Tactin beads (Figs 2B, S3C and S3D).

We conclude that RomY interacts separately with MglB and MglA-GTP. The observation that these interactions were only observed after cross-linking suggests that they are low affinity giving rise to transient complex formation. Because RomY stimulates MglA GTPase activity in an MglB-dependent manner, we conclude that the three proteins also form a complex in which all three proteins are present. Because RomY alone does not have MglA GAP activity even though the two proteins interact, we conclude that RomY stimulates MglB GAP activity.

To gain insights into how RomY may interact with MglA-GTP and the MglB homodimer, we generated a structural model of RomY using AlphaFold [41] and the ColabFold pipeline

[42]. Because the oligomeric state of RomY is not known, we modelled RomY as a monomer. In all five models generated, residues 7–90, which covers the N-terminal conserved region from residue 8–89 (S1A Fig), were predicted to fold into a globular domain with high accuracy based on Predicted Local Distance Difference Test (pLDDT) and predicted alignment error (pAE), while the remaining parts of RomY was modelled with lower accuracy (Figs 3A and S4A). Because the N-terminal conserved part of RomY extends from residue 8 to 89, from hereon we refer to residue 1–89 as the N-terminal domain of RomY.

To understand how RomY may interact with MglA, the MglB homodimer and the two proteins in parallel, we used AlphaFold-Multimer [43] to generate models of MglA:RomY, $(MglB)_2$:RomY and MglA:$(MglB)_2$:RomY complexes as well as of an MglA:$(MglB)_2$ complex. All five models of MglA:$(MglB)_2$ were predicted with high accuracy and are in overall agreement with the solved structure of MglA-GTPγS:$(MglB)_2$ [30,32] (S4A and S4B Fig) documenting the quality of the predictions and that AlphaFold-Multimer models MglA in the GTP-bound form in this complex.

For each of the five models of MglA:RomY, $(MglB)_2$:RomY and MglA:$(MglB)_2$:RomY complexes, we obtained high accuracy predictions based on pLDDT and pAE scores including residues 7–90 in RomY (S4A Fig). Therefore, we only considered the N-terminal domain of RomY in the structural models. In the MglA:RomY model, MglA had a structure similar to that of the solved structure of MglA-GTPγS [30,32] (S4C Fig), and, thus, AlphaFold-Multimer models MglA in the GTP bound form. Notably, the N-terminal domain of RomY associated with MglA close to the nucleotide-binding pocket (S4C Fig). In the $(MglB)_2$:RomY model, the MglB homodimer was similar to the solved structure [30,32], and the RomY N-terminal domain interacted asymmetrically with the two MglB monomers using a different surface than for the interaction with MglA (S4D Fig). Finally, in the model of all three proteins, the MglA: $(MglB)_2$ part was similar to the solved structure of the MglA-GTPγS:$(MglB)_2$ complex [30,32] (S4E Fig), documenting that also in this context AlphaFold-Multimer models MglA in the GTP-bound state. Importantly, the N-terminal domain of RomY interacted with MglA and one of the MglB monomers in the MglB homodimer, and was positioned close to the nucleotide-binding pocket of MglA (Fig 3B). Thus, the MglA:$(MglB)_2$:RomY model supports that all three proteins interact to form a complex in which they interact in all three pairwise directions and in which MglA is in the GTP-bound state. Also, this model agrees with the interactions detected in the pull-down experiments (Figs 2B, S3C and S3D). Moreover, this model suggests that the N-terminal domain of RomY has a key role in stimulating MglB GAP activity.

To test this structural model, we generated a RomY variant (RomY[N]) that was truncated for the C-terminal part of RomY and only included residue 1–89 (Figs 3A, 3B and S1A). In the *in vitro* GTPase assays, Strep-RomY[N] stimulated MglA GTPase activity in the presence of MglB, although at a slightly but significantly lower level than full-length Strep-RomY (Figs 2A and S3A). *In vivo*, a mutant synthesizing RomY[N] as the only RomY protein had a motility phenotype between WT and the Δ*romY* mutant (Figs 3C, S5A and S5B). Although neither purified Strep-RomY[N] nor RomY[N] synthesized *in vivo* were detectable in immunoblot analysis with α-RomY antibodies, these observations support that the N-terminal domain of RomY interacts with MglA-GTP as well as the MglB dimer and that the MglB GAP stimulating activity of RomY largely resides in this region of RomY.

## RomY is essential for sufficient MglB GAP activity *in vivo*

MglB alone has MglA GAP activity *in vitro*. However, the Δ*mglB* and Δ*romY* mutations cause similar motility defects *in vivo* suggesting that RomY is required for sufficient MglB GAP activity *in vivo*. Alternatively, MglB alone has GAP activity *in vivo*, but its concentration is too

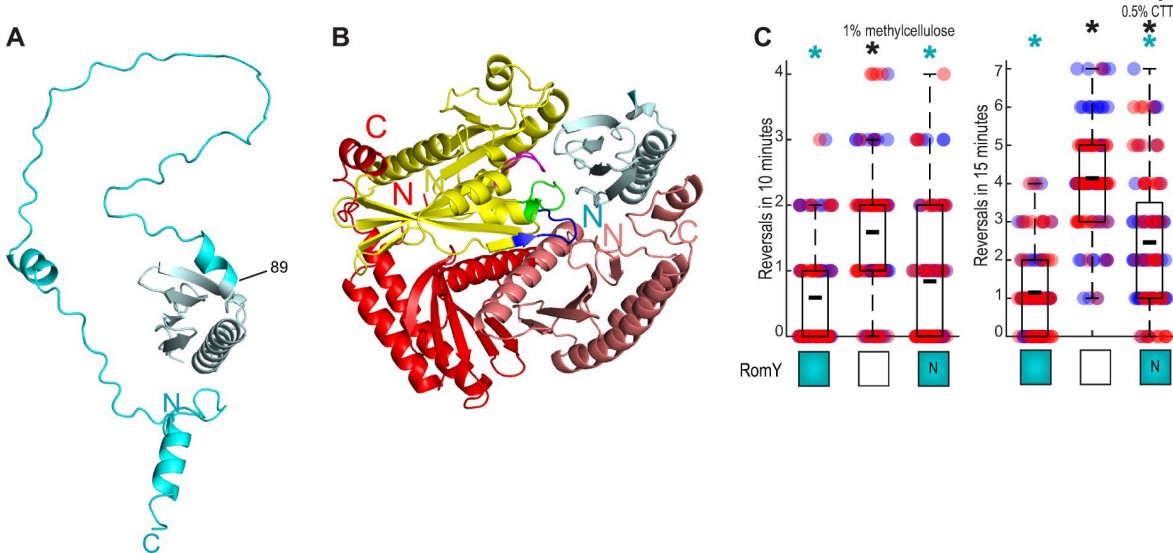

**Fig 3. The N-terminal domain of RomY has partial RomY activity. A.** AlphaFold model of RomY. RomY was modeled as a monomer. The N-terminal conserved region up to residue 89 is in teal and the remaining part in cyan. Model rank 1 is shown. **B.** AlphaFold-Multimer model of the MglA:(MglB)$_2$:RomY complex. The MglA monomer is in yellow and with the P-loop in purple, switch region-1 in blue and switch region-2 in green, the MglB homodimer in red, and the N-terminal domain of RomY in teal. **C.** The N-terminal domain of RomY has partial RomY activity. Reversals were tracked in single cells for T4P-dependent and gliding motility as in Fig 1D and 1E. Boxes below diagrams indicate the presence or absence of RomY as colored or white boxes, respectively. N indicates RomY$^N$. Individual data points from two independent experiment with each $n$ = 50 cells are plotted in red and blue. Boxplot is as in Fig 1D and 1E. * $P < 0.05$, two-sided Student's $t$-test with comparison to WT (black) and the $\Delta romY$ mutant (cyan).

low to stimulate MglA GTPase activity sufficiently. To resolve the importance of RomY for MglB GAP activity *in vivo*, we overexpressed MglB in the presence or absence of RomY. Cells with a low level of MglA-GTP are non-motile by gliding but move with WT speed and a reduced reversal frequency using the T4P-dependent motility system [27,29,40] (Cf. Fig 1D and 1E $\Delta romR$ and $\Delta romX$ mutants). Therefore, we used speed and reversal frequency in T4P-dependent motility as precise and sensitive readouts of GAP activity in these experiments.

MglB and RomY accumulated independently when expressed from their native loci (Fig 4A). MglB was ectopically overproduced using a vanillate-inducible promoter (P$_{van}$) in $\Delta mglB$ strains. In the absence of vanillate, MglB was not detectable in immunoblots; upon addition of 500 μM vanillate, MglB accumulated at an ~20-fold higher level than in WT and independently of RomY. The level of RomY was unaffected by the increased MglB accumulation (Fig 4A). In the absence of vanillate, the $\Delta mglB$/P$_{van\_}mglB$ strain containing RomY was similar to the $\Delta mglB$ strain and hyper-reversed (Figs 4B and S6). Importantly, in the presence of vanillate, cells of this strain moved with WT speed but had a reversal frequency significantly below that of WT (Figs 4B and S6), indicating a low MglA-GTP concentration. By contrast, in the absence of RomY, MglB overproduction did not affect the reversal frequency. In the inverse experiments, RomY was ectopically overproduced from a vanillate-inducible promoter in the absence or presence of MglB (Figs 4A and S6). RomY overproduction (~10-fold higher than the RomY level in WT) in the presence of MglB resulted in a reversal frequency significantly below that of WT. By contrast, RomY overexpression in the absence of MglB did not affect the reversal frequency.

We conclude that the MglB and RomY proteins accumulate independently of each other. In addition, neither MglB nor RomY alone, even when highly overproduced, is sufficient to stimulate MglA GTPase activity to WT levels. Rather MglB, even when overproduced, depends

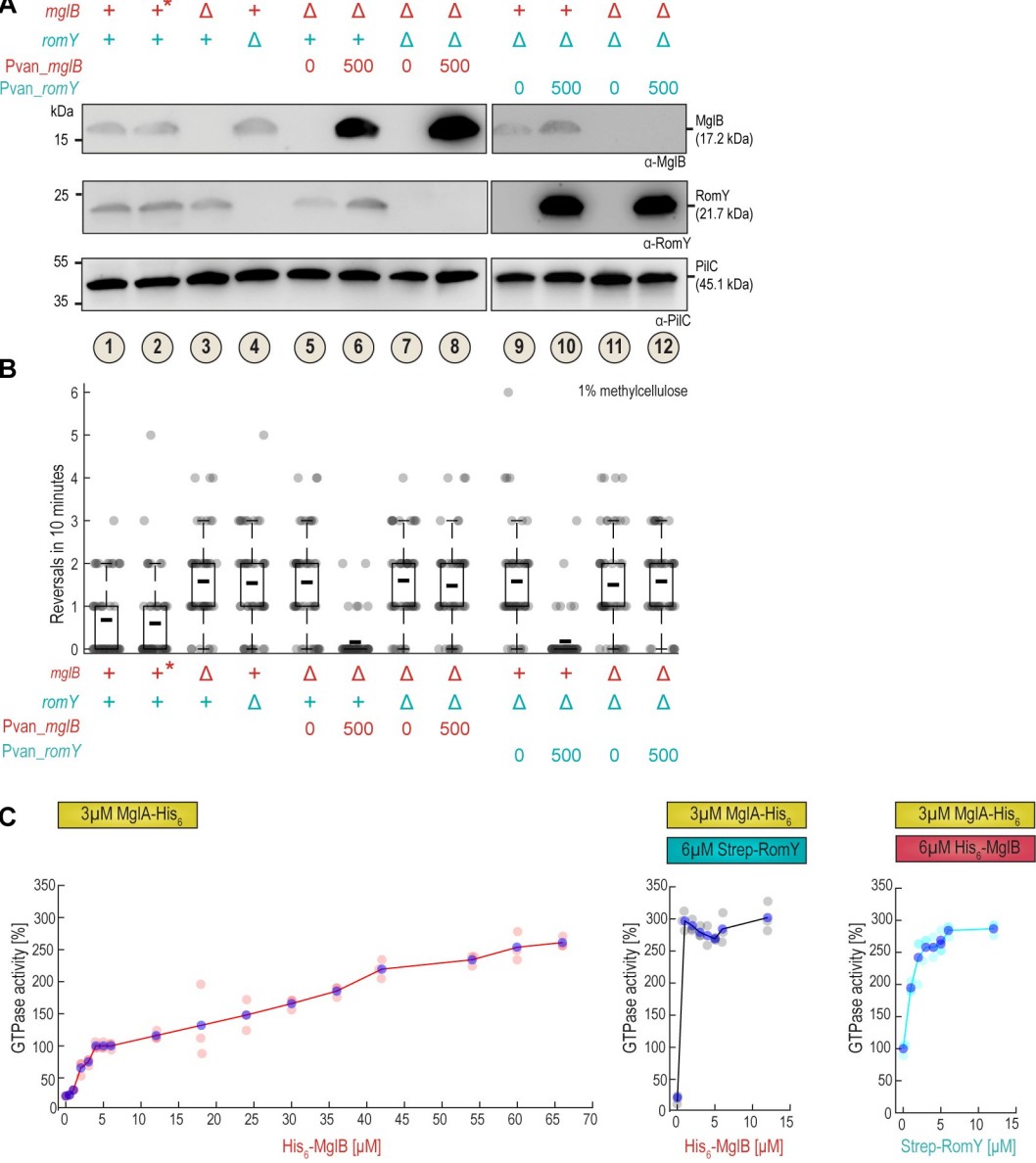

**Fig 4. RomY is essential for sufficient MglB sufficient GAP activity *in vivo*. A.** Analysis of MglB and RomY accumulation by immunoblot analysis in induction experiments. Strains of the indicated genotypes were grown in the presence and absence of 500μM vanillate for 3h as indicated. Cell lysates prepared from the same number of cells for each sample were separated by SDS–PAGE and probed sequentially with α-MglB, α-RomY and α-PilC (loading control) antibodies with stripping of the membrane before the second and third antibodies. In the legend, + indicates presence of WT gene, Δ in-frame deletion, 0/ 500μM vanillate concentration, and * the WT grown in the presence of 500μM vanillate. Samples 1–8 and 9–12 were separated on different SDS-PAGE gels that both contained samples 1–4 to enable comparisons of samples between different gels. **B.** Analysis of reversals in T4P-dependent motility upon overproduction of MglB or RomY. Cells were treated as in (**A**) and then T4P-dependent single cell motility analyzed. Legend is as in (**A**). Individual data points from a representative experiment with *n* = 50 cells are plotted in gray. Because the experiment relies on induction of gene expression, protein levels vary slightly between experiments, making the direct comparison between biological replicates difficult. Consequently, data from only one representative experiment is shown. Boxplots are as in Fig 1D. **C.** MglA GTPase activity measured as GTP turnover in the regenerative coupled enzyme assay. The activity in the presence of 3μM MglA-His$_6$, 6μM His$_6$-MglB (condition in Fig 2A) is set to 100%. Left panel, 3μM MglA-His$_6$ titrated with increasing concentrations of His$_6$-MglB; middle panel, 3μM MglA-His$_6$ and 6μM Strep-RomY titrated with increasing concentrations of His$_6$-MglB; right panel, 3μM MglA-His$_6$ and 6μM μM MglB-His$_6$ titrated with increasing concentrations of Strep-RomY. Individual data points from three independent experiments are indicated in light red, grey and cyan dots; blue dots indicate the mean.

on RomY for efficient GAP activity *in vivo*; similarly, RomY only results in GAP activity in the presence of MglB *in vivo*. These observations also support that RomY stimulates MglB activity and is essential for sufficient MglB GAP activity *in vivo*. Notably, overproduction of either MglB or RomY in the presence of WT levels of RomY or MglB, respectively, results in increased GAP activity compared to WT. These observations corroborate that the rate-limiting step for MglB/RomY GAP activity is complex formation, i.e. when one of the proteins is over-produced in the presence of the other, then more of the MglB/RomY complex is formed, thus, also supporting that the MglB/RomY interaction is low affinity.

To analyze the effect of varying MglB and RomY concentrations on MglA GTPase activity *in vitro*, we used the regenerative coupled GTPase assay. At 3μM MglA-His$_6$, GTPase activity increased linearly with increasing His$_6$-MglB slowly reaching saturation at ~6μM His$_6$-MglB; fur-ther increasing the His$_6$-MglB concentration >10-fold to 65μM only resulted in a ~2.5-fold increase in GTPase activity (Fig 4C, left). When 3μM MglA-His$_6$ was titrated with increasing His$_6$-MglB concentrations in the presence of 6μM Strep-RomY, 1μM His$_6$-MglB was sufficient for the reaction to reach saturation (Fig 4C, middle), and at saturation, the MglA-His$_6$ GTPase activity under these conditions was slightly higher than in the presence of 65μM His$_6$-MglB alone (Fig 4C, middle). In the inverse experiment, we titrated increasing concentrations of Strep-RomY against 3μM MglA-His$_6$ and 6μM His$_6$-MglB. Saturation of the GTPase reaction was reached at 3–6μM Strep-RomY (Fig 4C, right), and at saturation, also under these conditions, the MglA-His$_6$ GTPase activity was slightly higher than in the presence of 65μM His$_6$-MglB alone (Fig 4C, right).

In conclusion, under the conditions of the GTPase assay, His$_6$-MglB *in vitro* stimulates MglA-His$_6$ GTPase activity more efficiently in the presence of Strep-RomY than can be achieved by simply increasing the His$_6$-MglB concentration in the absence of Strep-RomY. Thus, overall, these *in vitro* observations parallel the *in vivo* observations in which ~20-fold MglB overproduction in the absence of RomY was not sufficient to reach the GAP activity observed in WT. Of note, several factors make a direct comparison between the *in vitro* and *in vivo* data difficult. First, in the *in vitro* system, GAP activity is measured in the absence of GEF activity; however, *in vivo*, both activities are present. Therefore, overproduced MglB *in vivo* may have GAP activity, but it is too low to outcompete the GEF activity of the RomR/RomX complex (see Discussion). Second, the cellular and polar concentrations of MglA, MglB and RomY are not known and, therefore, it is not known where on the titration curves the relevant physiologically concentrations are. Third, His$_6$-MglB alone at very high concentrations *in vitro* stimulated MglA-His$_6$ GTPase activity almost to the same level as His$_6$-MglB together with Strep-RomY. Nevertheless, the ~20-fold overproduction of MglB in the absence of RomY was not sufficient to reach the MglB/RomY GAP activity in WT. Thus, it remains a possibility that if MglB could be overproduced to even higher levels, then MglB alone would be sufficient to display detectable GAP activity *in vivo*.

## RomY localizes dynamically to the lagging cell pole in an MglB-dependent manner

Our data are consistent with RomY stimulating MglB GAP activity *in vitro* and *in vivo*. To resolve if RomY contributes to the spatial regulation of MglB GAP activity *in vivo*, we deter-mined RomY localization using an ectopically expressed, active RomY-YFP fusion (S7A and S7B Fig). By snapshot analysis, RomY-YFP localized in a highly asymmetric pattern with 81% of cells having unipolar or asymmetric bipolar localization (Fig 5A). In moving cells, the large cluster localized highly asymmetrically to the lagging cell pole (Fig 5B; see also below). More-over, RomY-YFP localization was dynamic, and after a reversal, RomY-YFP localized to the new lagging pole (Fig 5B).

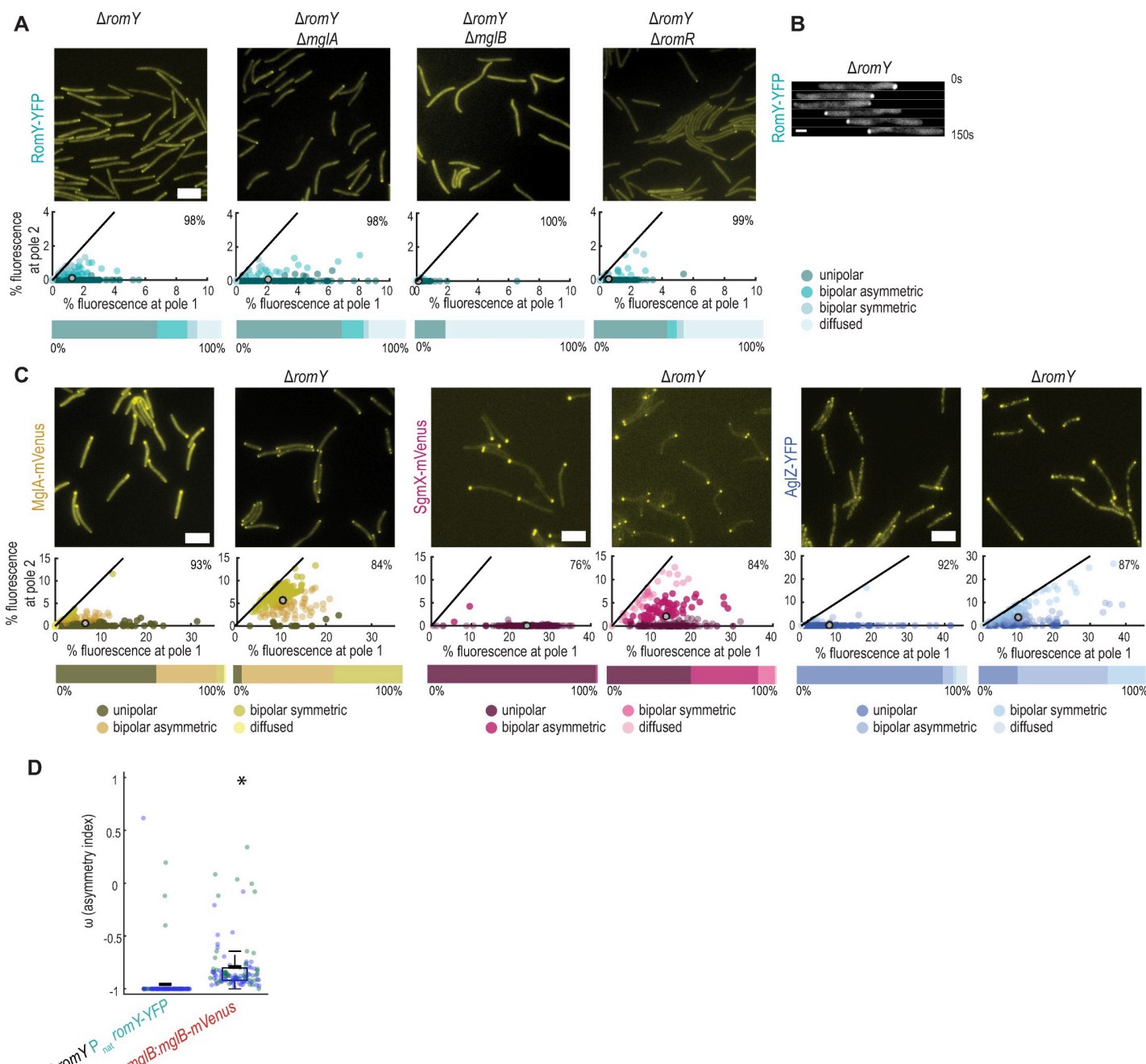

**Fig 5. RomY localizes dynamically to the lagging cell pole. A.** RomY-YFP localization by epi-fluorescence microscopy. In the scatter plot, the percentage of total fluorescence at pole 2 is plotted against the percentage of total fluorescence at pole 1 for all cells with polar cluster(s). Pole 1 is per definition the pole with the highest fluorescence. Individual cells are color-coded according to its localization pattern. Black lines are symmetry lines, grey spots show the mean and numbers in the upper right corner the mean percentage of total fluorescence in the cytoplasm. Horizontal bars below show the percentage of cells with a polar localization pattern and diffuse localization according to the color code. $n$ = 200 cells in all strains. Scale bar, 5 μm. **B.** RomY-YFP is dynamically localized to the lagging pole. Cells were imaged by time-lapse epi-fluorescence microscopy every 30s. Scale bar, 1μm. **C.** MglA-mVenus, Sgmx-mVenus and AglZ-YFP localization in the absence of RomY. Cells were imaged by epi-fluorescence microscopy, scatter plots and percentage of cells with a particular localization pattern were determined as in **A**. $n$ = 200 cells for all strains. Scale bar, 5 μm. **D.** Comparison of RomY-YFP and MglB-mVenus asymmetry in moving cells. Cells were imaged by time-lapse epi-fluorescence microscopy every 30s. An asymmetry index (ω) was calculated for cells that moved for three or more successive frames without reversing and excluding the first frame after a reversal and the last frame before a reversal (see Materials and Methods). ω = -1, unipolar localization at the lagging pole, ω = +1, unipolar localization at the leading pole, and ω = 0, bipolar symmetric localization. Individual data points from two independent experiments (27/33 cells and 48/54 data points for RomY-YFP, and 7/11 cells and 53/53 data points for MglB-mVenus) are plotted in blue and green. Boxplot is as in Fig 1D and 1E. * $P < 0.005$, two-sided Student's $t$-test.

To determine how RomY-YFP is targeted to the lagging pole, we interrogated RomY-YFP localization in the absence of MglA, MglB or RomR, which we used as a proxy for the RomR/RomX complex (Fig 5A). RomY-YFP accumulated as in WT in the absence of each of these proteins (S7B Fig). In the absence of MglA, the total polar RomY-YFP signal was as in WT and the protein was slightly more unipolar (Fig 5A). In the absence of MglB, RomY-YFP polar localization was strongly reduced, and 81% of cells had no polar signals and the remaining signals were of low intensity (Fig 5A). In the absence of RomR, RomY-YFP polar localization was also strongly reduced, and only 54% of cells still had a weak polar signal (Fig 5A). Of note, polar localization of MglB, MglA and RomX is strongly reduced or even abolished in the absence of RomR [29,33,35,36], but MglA, RomR and RomX are still polarly localized in the absence of MglB [36]. In agreement with the direct interaction between RomY and MglB, we, therefore, conclude that MglB is the primary polar targeting determinant of RomY-YFP, and that the reduced polar localization of RomY-YFP in the absence of RomR is caused by an indirect effect of RomR on MglB localization. Moreover, our data support that MglA has at most a minor role in polar RomY localization. We note that in the absence of MglB or RomR, there is still some residual unipolar RomY-YFP localization. MglA is more bipolarly localized in the absence of MglB, supporting that MglA does not bring about this residual polar localization and that yet to be identified factor(s) may have a role in RomY polar localization.

To determine the effect of RomY on the proteins of the polarity module, we focused on MglA because this protein generates the output of this module. We also analyzed the two MglA-GTP effectors SgmX and AglZ that localize to the leading pole in an MglA-dependent manner to stimulate the formation of T4P [27,28] and assembly of the Agl/Glt complexes [15], respectively and, thus, provide a functional readout of the state of the polarity module. As previously shown, MglA-mVenus localized in a highly asymmetric pattern in WT (Figs 5C and S7C). Notably, in the absence of RomY, polar localization of MglA-mVenus was increased and switched toward more bipolar symmetric (Fig 5C). Thus, RomY, as previously observed for MglB, is required to exclude MglA-GTP from the lagging pole. In agreement with these observations and that the ΔromY mutant hyper-reverses in a Frz-independent manner, SgmX-mVenus and AglZ-YFP were shifted from strongly unipolar toward bipolar symmetric in the absence of RomY (Figs 5C, S7D and S7E). Importantly, in the ΔmglB mutant, localization of SgmX and AglZ is also shifted toward bipolar symmetric [27,35].

Altogether, we conclude that MglB is the primary determinant of polar RomY localization, that RomY stimulates MglB GAP activity at the lagging pole, and, together with MglB, RomY is required to exclude MglA-GTP from this pole.

### RomY specifically stimulates MglB GAP activity at the lagging cell pole

Because MglB is bipolarly asymmetrically localized, we reasoned that for RomY to only stimulate MglB activity at the lagging pole and, thus, spatially confine MglB/RomY GAP activity to this pole, RomY would have to be more asymmetrically localized to the lagging pole than MglB. To this end, we determined the polar asymmetry of RomY-YFP and an active MglB-mVenus fusion (S7A and S7F Fig) in moving cells. As shown in Fig 5D, RomY-YFP was almost exclusively unipolar, while MglB-mVenus was almost exclusively bipolar asymmetric. Thus, RomY is significantly more asymmetrically localized to the lagging pole than MglB supporting that RomY only stimulates MglB GAP activity at this pole.

### Discussion

In the rod-shaped *M. xanthus* cells, the activity of the small GTPase MglA is spatially restricted to the leading cell pole by the joint action of its cognate GEF and GAP. This spatial regulation

ensures that the two motility systems only assemble at the leading cell pole and is, thus, critical for directed motility. The RomR/RomX GEF and the MglB GAP localize similarly to the two poles but with GEF and GAP activity dominating at the leading and lagging cell pole, respectively. How this spatial separation of these two activities is brought about has remained unknown. Here, we report the identification of the previously uncharacterized RomY protein and demonstrate that it is an integral part of the polarity module. Specifically, RomY interacts with MglB to form a bipartite, low-affinity MglB/RomY complex and stimulates MglB GAP activity. RomY almost exclusively localizes to the lagging cell pole with its high MglB concentration, thereby stimulating MglB GAP activity at this pole. Conversely, RomY largely does not localize to the leading cell pole with its low MglB concentration and, therefore, does not significantly stimulate MglB GAP activity at this pole. Altogether, our *in vivo* and *in vitro* data suggest that RomY by engaging in the formation of the bipartite MglB/RomY GAP complex is essential to reach the high GAP activity that allows the MglB/RomY GAP activity to outcompete and dominate over the RomR/RomX GEF activity at this pole.

*In vitro* MglB alone has MglA GAP activity, while RomY alone does not. However, RomY stimulates MglA GTPase activity *in vitro* in the presence of MglB. *In vitro* RomY interacts independently and with low affinity with MglB and MglA-GTP in pull-down experiments. Because RomY interacts separately with MglB and MglA-GTP but does not have GAP activity on its own, we conclude that RomY stimulates MglB activity. Canonical GAPs of Ras-like GTPases supply either an arginine finger or an asparagine thumb to complete the active site of the GTPase [11,12]. By contrast, the MglB dimer interacts asymmetrically with MglA-GTP and brings about the repositioning of amino acid residues in MglA to generate the active site for GTP hydrolysis [30–32]. A high accuracy AlphaFold-Multimer based structural model of the MglA:(MglB)$_2$:RomY complex supports that the N-terminal domain of RomY (RomY$^N$) interacts with one of the MglB monomers in the MglB homodimer as well as with MglA close to the nucleotide-binding pocket. Consistently, RomY$^N$ stimulates MglB GAP activity partially *in vitro* and has partial RomY activity *in vivo*. Continued biochemical work and structural studies will be required to determine the affinity of RomY for the MglA/MglB complex and the stoichiometry of the MglA-GTP/MglB/RomY complex as well as to decipher the exact mechanism by which RomY binds to the MglA/MglB complex and stimulates MglB GAP activity.

We only observed the interactions between RomY and MglB and MglA-GTP after protein cross-linking, suggesting that the MglB/RomY and MglA-GTP/RomY complexes are low affinity and transient. Overproduction of MglB *in vivo* in the presence of WT levels of RomY caused an increase in MglB/RomY GAP activity. Similarly, overproduction of RomY in the presence of WT levels of MglB caused an increase in MglB/RomY GAP activity. Altogether, these observations support that the rate-limiting step for MglB/RomY GAP activity in WT is the formation of the MglB/RomY complex, corroborating that the MglB/RomY complex is a low affinity. *In vivo* RomY localizes almost exclusively to the lagging cell pole and this localization depends strongly on MglB. By contrast, MglB localizes in a bipolar asymmetric pattern with the highest concentration at the lagging pole. Building on these observations, we suggest that RomY localizes almost exclusively to the lagging cell pole due to the high concentration of MglB at this pole. By contrast, RomY essentially does not localize to the leading pole because the concentration of MglB would be too low at this pole to support MglB/RomY complex formation. In principle, RomY could be recruited to the leading pole by MglA-GTP; however, we observed that MglA-GTP does not appear to play a role in the polar recruitment of RomY. As a consequence of this RomY localization pattern, the highly active MglB/RomY GAP is almost exclusively present at the lagging pole where both proteins are essential for excluding MglA-GTP from this pole. In principle, RomY could also help to exclude MglA-GTP from the

lagging pole by inhibiting the RomR/RomX GEF at this pole; however, the epistasis experiments (Fig 1D and 1E) do not support this scenario.

*In vitro* MglB alone has GAP activity and this activity is stimulated by RomY. However, the Δ*romY* mutant phenocopies the Δ*mglB* mutant supporting that RomY is essential for sufficient MglB GAP activity to regulate polarity *in vivo*. Even when MglB was overproduced 20-fold in the absence of RomY, its GAP activity was not sufficient to reach the level in WT. Our data do not allow us to distinguish whether RomY is essential for MglB GAP activity *in vivo* or for *sufficiently high* MglB GAP activity *in vivo* for MglB to regulate polarity. We speculate that RomY *in vivo*, as observed *in vitro*, boosts MglB GAP activity to a sufficiently high level to outcompete RomR/RomX GEF activity at the lagging pole (see below).

The RomR/RomX GEF and MglB GAP are both arranged intracellularly with a high concentration at the lagging pole and a low concentration at the leading pole. Nevertheless, GEF activity dominates at the leading and GAP activity at the lagging cell pole. It has been argued that this localization pattern is ideal to allow stable as well as switchable polarity and, thus, reflects a trade-off between maintaining stable polarity with unidirectional motility between reversals and sensitivity to Frz signaling with an inversion of polarity and cellular reversals [36]. Briefly, with this paradoxical localization pattern, the RomR/RomX GEF complex at the lagging pole would be ideally positioned to recruit MglA during reversals [36]. However, the "price" that cells pay for this design is the need for a mechanism to separate the GEF and GAP activities spatially. It has remained enigmatic how the spatial regulation of the GEF and GAP is brought about. The data presented here suggest that RomY is an elegant solution to this problem. Specifically, because the MglB/RomY complex is low-affinity, it is formed almost exclusively at the lagging pole with the high MglB concentration but not at the leading pole with the low MglB concentration. In this way, MglB GAP activity is almost exclusively, if not only, stimulated at the lagging pole.

Based on the data reported here, we suggest a revised model for the regulation of front-rear polarity in *M. xanthus*. In this model, antagonistically acting GEF and GAP complexes are present at both poles. At the leading pole, the RomR/RomX GEF complex is active, while the MglB(/RomY) GAP has low activity because RomY is largely, if not completely, absent. As a consequence, RomR/RomX GEF activity dominates and outcompetes MglB(/RomY) GAP activity resulting in MglA activation and recruitment to the leading pole. By contrast, at the lagging pole, the two antagonistically acting RomR/RomX GEF and MglB/RomY GAP complexes are both present and active. Our genetic and cell biology data support that MglB/RomY GAP activity dominates and outcompetes the RomR/RomX activity at this pole. In other words, RomY complexed to MglB with the formation of the more highly active MglB/RomY GAP complex at the lagging pole helps to push the competition between the two antagonistically acting complexes in favor of the GAP activity. As a result, MglA-GTP is inactivated (i.e. MglA GTPase activity is activated) by the MglB/RomY GAP complex and MglA does not accumulate at this pole (Fig 6). Thus, the key to the spatial regulation of the antagonistic GEF and GAP activities is the low affinity of RomY for MglB that restricts the high MglB/RomY GAP activity to the lagging pole. In this model, the RomR/RomX GEF and MglB/RomY GAP complexes are both active at the lagging pole potentially giving rise to futile cycles of GEF and GAP activity with GDP-for-GTP exchange and GTP hydrolysis by MglA. Currently, there is no evidence supporting that the two activities are cross-regulated. In future experiments, it will, therefore, be important to determine the overall cellular and polar concentrations of the involved proteins to estimate the magnitude of these futile cycles and also to relate the measured GEF and GAP activities *in vitro* to the *in vivo* protein concentrations.

In eukaryotes, Rho GTPases are key regulators of motility and polarity and their activity is spatially confined to distinct intracellular locations. In some cases, this confinement has been

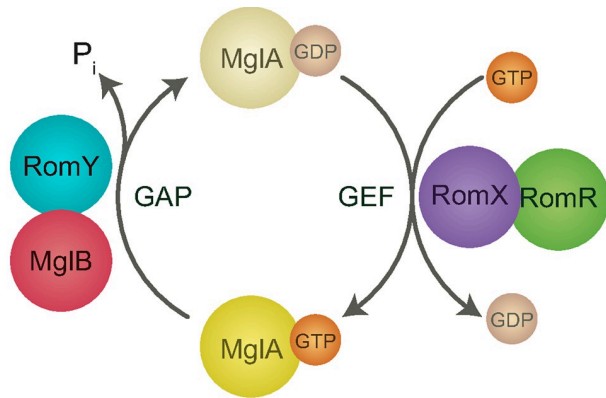

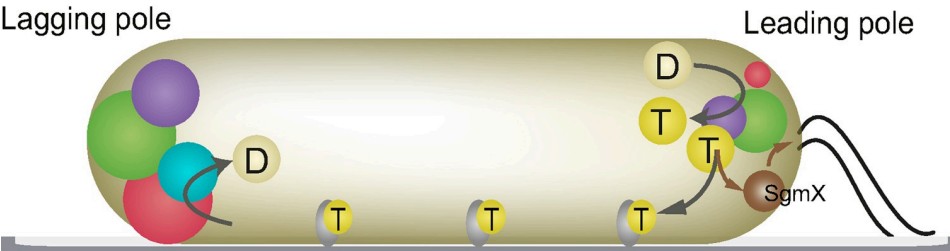

**Fig 6. Model for front–rear polarity in *M. xanthus*.** Upper panel, MglA GTPase cycle. The MglB/RomY complex is shown to indicate that both proteins interact with MglA-GTP and the RomR/RomX complex to indicate that only RomX interacts with MglA. Lower panel, localization of MglA-GTP, MglB, RomY, RomR, RomX and SgmX in a cell with T4P at the leading pole. Color code as in the upper panel, except that yellow circles labelled D and T represent MglA-GDP and MglA-GTP, respectively. SgmX is in brown and with the brown arrows indicating its recruitment by MglA-GTP and stimulation of T4P formation. The dark grey arrow indicates stimulation of assembly of the Agl/Glt complexes (light grey) and the incorporation of MglA-GTP into these complexes. Circle sizes indicate the amount of protein at a pole.

shown to rely on spatially separated GEF and GAP activities. For instance, in *Drosophila* epithelial cells, Cdc42 colocalizes with its cognate GEFs at the apical membrane while the cognate GAP is at the lateral membrane and assists in restricting Cdc42 activity to the apical membrane [44]. Thus, the design principles underlying polarity are overall similar in these systems and *M. xanthus*. However, in *M. xanthus*, polarity can be inverted, while it is stably maintained in epithelial cells. As mentioned, it has been suggested that the special arrangement with the RomR/RomX GEF in a "waiting position" at the lagging pole is key to this switchability [36]. Because Ras-like GTPases are also involved in regulating dynamic polarity in eukaryotes [5,8], we speculate that polarity systems with a design similar to the *M. xanthus* system may underlie the regulation of dynamic polarity in eukaryotic cells.

Small Ras-like GTPases and Roadblock proteins are present in all three domains of life, and it has been suggested that they were present in the last universal common ancestor [37,45–47]. Interestingly, proteins containing a Roadblock domain or the structurally related Longin domain, which might have evolved from the Roadblock domain [48], often form heteromeric complexes with GEF or GAP activity. For instance, the Ragulator complex has Rag GEF activity [49–52], the GATOR1 and FLCN/FNIP complexes Rag GAP activity [53–55], and the Mon1-Ccz1 and TRAPP-II complexes have Rab7 GEF [56] and Rab1 GEF [57] activity, respectively. Thus, the finding that MglB functions in a complex with RomY follows this theme and adds the MglB/RomY complex to the list of heteromeric Roadblock domain-containing

complexes important for regulating small GTPases. These observations also support the idea that cognate GTPase/Roadblock pairs could represent minimal, ancestral pairs of a GTPase and its regulator. During evolution, Roadblock domain-containing proteins would then have become incorporated into more complex GEFs and GAPs to regulate GTPase activity.

## Materials and methods

### Cell growth and construction of strains

DK1622 was used as the WT *M. xanthus* strain and all strains are derivatives of DK1622. *M. xanthus* strains used are listed in S2 Table. Plasmids are listed in S3 Table. In-frame deletions were generated as described [58]. *M. xanthus* was grown at 32˚C in 1% casitone (CTT) broth [59] or on 1.5% agar supplemented with 1% CTT and kanamycin (50μg/ml) or oxytetracycline (10μg/ml) if appropriate. Plasmids were integrated by site specific recombination into the Mx8 *attB* site or by homologous recombination at the native site. All in-frame deletions and plasmid integrations were verified by PCR. Primers used are listed in S4 Table. Plasmids were propagated in *Escherichia coli* TOP10 (F⁻, *mcrA*, Δ(*mrr-hsd*RMS-*mcr*BC), φ80*lacZ*ΔM15, Δ*lac*X74, *deo*R, *rec*A1, *ara*D139, Δ(*ara-leu*)7679, *gal*U, *gal*K, *rps*L, *end*A1, *nup*G) unless otherwise stated. *E. coli* cells were grown in LB or on plates containing LB supplemented with 1.5% agar at 37˚C with added antibiotics if appropriate [60]. All DNA fragments generated by PCR were verified by sequencing.

### Motility assays and determination of reversal frequency

Population-based motility assays were done as described [38]. Briefly, *M. xanthus* cells from exponentially growing cultures were harvested at 4000× *g* for 10 min at room temperature (RT) and resuspended in 1% CTT to a calculated density of $7×10^9$ cells ml⁻¹. 5μL aliquots of cell suspensions were placed on 0.5% agar plates supplemented with 0.5% CTT for T4P-dependent motility and 1.5% agar plates supplemented with 0.5% CTT for gliding motility and incubated at 32˚C. After 24h, colony edges were visualized using a Leica M205FA stereomicroscope and imaged using a Hamamatsu ORCA-flash V2 Digital CMOS camera (Hamamatsu Photonics). For higher magnifications of cells at colony edges on 1.5% agar, cells were visualized using a Leica DMi8 inverted microscope and imaged with a Leica DFC9000 GT camera. Individual cells were tracked as described [29]. Briefly, for T4P-dependent motility, 5μL of exponentially growing cultures were spotted into a 24-well polystyrene plate (Falcon). After 10min at RT, cells were covered with 500μL of 1% methylcellulose in MMC buffer (10mM MOPS (3-(*N*-morpholino)propanesulfonic acid) pH 7.6, 4mM MgSO$_4$, 2mM CaCl$_2$), and incubated at RT for 30min. Subsequently, cells were visualized for 10min at 20sec intervals at RT using a Leica DMi8 inverted microscope and a Leica DFC9000 GT camera. Individual cells were tracked using Metamorph 7.5 (Molecular Devices) and ImageJ 1.52b [61] and then the speed of individual cells per 20sec interval as well as the number of reversals per cell per 10min calculated. For gliding, 5μL of exponentially growing cultures were placed on 1.5% agar plates supplemented with 0.5% CTT, covered by a cover slide and incubated at 32˚C. After 4 to 6h, cells were observed for 15min at 30sec intervals at RT as described above and then the fraction of moving cells, speed per 30sec interval as well as the number of reversals per 15min calculated.

For experiment with vanillate, cells were diluted to the same optical density (OD) at 550nm of 0.2, grown for 30min at 32˚C in suspension culture, and then vanillate was added to a final concentration of 500μM. Subsequently, cells were grown 3h at 32˚C before cells were spotted into a 24-well polystyrene plate (Falcon). After 10min at RT, cells were covered with 500μL of 1% methylcellulose in MMC buffer supplemented with 500μM vanillate, and incubated at RT

for 30min. Subsequently, cells were visualized for 10min at 20sec intervals at RT as described. Control cultures without vanillate were treated similarly.

## Fluorescence microscopy

Epifluorescence microscopy was done as described [29]. Briefly, *M. xanthus* cells were placed on a thin 1.5% agar pad buffered with TPM buffer (10mM Tris-HCl pH 8.0, 1mM potassium phosphate buffer pH 7.6, 8mM $MgSO_4$) on a glass slide and immediately covered with a coverslip. After 30min at 32˚C, cells were visualized using a Leica DMi8 microscope and imaged with Hamamatsu ORCA-flash V2 Digital CMOS camera. Cells in phase contrast images were automatically detected using Oufti [62]. Fluorescence signals in segmented cells were identified and analyzed using a custom-made Matlab v2016b (MathWorks) script [29]. Briefly, polar clusters were identified when they had an average fluorescence two STDEV above the average cytoplasmic fluorescence and a size of three or more pixels. For each cell with polar clusters, an asymmetry index ($\omega$) was calculated as

$$\omega = \frac{total\ fluorescence\ at\ pole\ 1 - total\ fluorescence\ at\ pole\ 2}{total\ fluorescence\ at\ pole\ 1 + total\ fluorescence\ at\ pole\ 2}$$

By definition, pole 1 is the pole with the highest fluorescence. $\omega$ varies between 0 (bipolar symmetric localization) and 1 (unipolar localization). The localization patterns were binned from the $\omega$ values as follows: unipolar ($\omega > 0.9$), bipolar asymmetric ($0.9 > \omega > 0.2$) and bipolar symmetric ($\omega < 0.2$). Diffuse localization was determined when no polar signal was detected.

For time-lapse epifluorescence microscopy, cells were prepared as described. Time-lapse recordings were made for 15min with images recorded every 30sec. Data were processed with Metamorph 7.5 and ImageJ 1.52b. Cells in phase contrast images were automatically detected using Oufti. Fluorescence signals in segmented cells were identified and analyzed using a custom-made Matlab script. Briefly, polar clusters were identified when they had an average fluorescence two STDEV above the average cytoplasmic fluorescence, an average fluorescence two-fold higher than the average the cytoplasmic fluorescence, and a size of three or more pixels. A custom-made Matlab script was used to track cells, detect reversals, leading and lagging cell poles, and to plot the data.

## Immunoblot analysis

Immunoblots were done as described [60]. Rabbit polyclonal antibodies α-MglA [23], α-MglB [23], α-PilC [20] and α-RomY antibodies were used together with goat anti-rabbit immunoglobulin G conjugated with horseradish peroxidase (Sigma) as secondary antibody. Monoclonal mouse anti-polyHistidine antibodies conjugated with peroxidase (Sigma) were used to detect $His_6$ tagged proteins. To generate rabbit, polyclonal α-RomY antibodies, purified $His_6$-RomY was used to immunize rabbit as described [60]. Blots were developed by using Luminata Crescendo Western HRP Substrate (Millipore) and visualized using a LAS-4000 luminescent image analyzer (Fujifilm).

## Protein purification

All proteins were expressed in *E. coli* Rosetta 2(DE3) (F- *ompT hsdS*$_B$($r_B^-$ $m_B^-$) *gal dcm* (DE3 pRARE2) at 18˚C or 37˚C. To purify $His_6$-tagged proteins, Ni-NTA affinity purification was used. Briefly, cells were washed in buffer A (50mM Tris pH 7.5, 150mM NaCl, 10mM imidazole, 5%glycerol, 5mM $MgCl_2$) and resuspended in lysis buffer A (50 ml of wash buffer A

supplemented with 1mM DTT, 100µg mL$^{-1}$ phenylmethylsulfonyl fluoride (PMSF), 10U mL$^{-1}$ DNase 1 and protease inhibitors–Complete Protease Inhibitor Cocktail Tablet (Roche)). Cells were lysed by sonication, cell debris removed by centrifugation (48000× $g$, 4°C, 30min), and cell lysate filtered through 0.45 µm Polysulfone filter (Filtropur S 0.45, Sarstedt). The cleared cell lysate was loaded onto a 5mL HiTrap Chelating HP column (GE Healthcare) preloaded with NiSO$_4$ as described by the manufacturer and equilibrated in buffer A. The column was washed with 20 column volumes of buffer A supplemented with 20mM imidazole. Proteins were eluted with buffer A using a linear imidazole gradient from 20-500mM. Fractions containing purified MglA-His$_6$ or His$_6$-MglB proteins were combined and loaded onto a HiLoad 16/600 Superdex 75 pg (GE Healthcare) gel filtration column that was equilibrated with buffer A without imidazole for use in GTPase assays or buffer C (20mM HEPES (4-(2-hydroxyethyl)-1-piperazineethanesulfonic acid) pH 8.0, 150mM NaCl, 5mM MgCl$_2$) for use in pull-down experiments. Fractions containing His$_6$-MalE were combined and loaded on a HiLoad 16/600 Superdex 200 pg (GE Healthcare) column equilibrated with buffer C. Fractions containing His$_6$-tagged proteins were pooled, frozen in liquid nitrogen and stored at -80°C.

To purify Strep-RomY and Strep-RomY$^N$, biotin affinity purification was used. Briefly, cells were washed in buffer D (100mM Tris pH 8.0, 150mM NaCl, 1mM EDTA, 1mM DTT) and resuspended in lysis buffer D (50 ml of buffer D supplemented with 100µg mL$^{-1}$ PMSF, 10U mL$^{-1}$ DNase 1 and protease inhibitors–Complete Protease Inhibitor Cocktail Tablet (Roche)). Cells were lysed and cleared lysate prepared as described and loaded onto a 5 mL Strep-Trap HP column (GE Healthcare), equilibrated with buffer D. The column was washed with 20 column volumes of buffer D. Protein was eluted with buffer E (150mM Tris pH 8.0, 150mM NaCl, 1mM EDTA, 2.5mM Desthiobiotin). Elution fractions containing Strep-RomY or Strep-RomY$^N$ were loaded onto a a HiLoad 16/600 Superdex 200 pg (GE Healthcare) gel filtration column that was equilibrated with buffer A without imidazole for use in GTPase assays or buffer C for use in pull-down experiments. Fractions with Strep-RomY or StrepRomY$^N$ were pooled, frozen in liquid nitrogen and stored at -80°C.

## GTPase assays

GTP-hydrolysis by MglA-His$_6$ was measured using a continuous, regenerative coupled GTPase assay [63] or by measuring released inorganic phosphate (P$_i$) after GTP hydrolysis using a malachite green assay [64]. The continuous, regenerative coupled GTPase assay was performed in buffer F (50mM Tris pH 7.5, 150mM NaCl, 5% glycerol, 1mM DTT, 7.5mM MgCl$_2$) supplemented with 495µM NADH (Sigma), 2mM phosphoenolpyruvate (Sigma), 18-30U mL$^{-1}$ pyruvate kinase (Sigma) and 27–42 U mL$^{-1}$ lactate dehydrogenase (Sigma). MglA--His$_6$ (final concentration: 10µM) was pre-loaded with GTP (final concentration: 3.3mM) for 30min at RT in buffer F. In parallel, His$_6$-MglB, Strep-RomY, Strep-RomY$^N$ or equimolar amount of His$_6$-MglB and Strep-RomY/Strep-RomY$^N$ (final concentrations of all proteins: 8.6 µM) were preincubated for 10min at RT in buffer F. Reactions were started in a 96-well plate (Greiner Bio-One) by adding His$_6$-MglB and/or Strep-RomY/Strep-RomY$^N$ to the MglA/GTP mixture. Final concentrations in these reactions: MglA-His$_6$: 3µM, His$_6$-MglB: 6µM, Strep-RomY/Strep-RomY$^N$: 6µM, GTP: 1mM. Absorption was measured at 340nm for 60min at 37°C with an Infinite M200 Pro plate-reader (Tecan) and the amount of hydrolyzed GTP per h per molecule of MglA-His$_6$ calculated. For each reaction, background subtracted GTPase activity was calculated as the mean of three technical replicates. In the malachite green assay, released P$_i$ during GTP hydrolysis was measured in buffer F. Proteins were used in concentrations and preincubated as described. GTPase reactions were performed in 96-well plates (Greiner Bio-One) at 37°C and started by adding His$_6$-MglB and/or Strep-RomY/Strep-

RomY[N] to the MglA/GTP mixture. Final concentrations as described. After 1h, reactions were stopped and the colour developed according to the manufacturer's manual (BioLegend) and absorption at 590nm measured using an Infinite M200 Pro plate-reader (Tecan). Subsequently, released $P_i$ was calculated from a standard curve, and the amount of released $P_i$ per h per MglA-His$_6$ molecule calculated.

## Pull-down experiments

In all experiments involving MglA-His$_6$, MglA-His$_6$ was preloaded with GTP or GDP (44.4μM protein, 22.2mM GTP/GDP) for 30min at RT in buffer C. Subsequently, equimolar amounts of Strep-RomY and MglA-His$_6$, His$_6$-MglB or His$_6$-MalE were incubated for 30min RT in buffer C. Final concentrations: MglA-His$_6$, His$_6$-MglB, His$_6$-MalE, Strep-RomY: 20μM, GTP/GDP 10mM. Where indicated, DSP was added to a final concentration of 200μM for 5min at RT. Next, all reactions were quenched with Tris pH 7.6 added to a final concentration of 100mM and incubated for 15min at RT. Subsequently, 20μl of Strep-Tactin coated magnetic beads (MagStrep 'type3' XT beads (IBA-Lifesciences)) previously equilibrated with buffer C were added and samples incubated for 30min RT. The beads were washed 10 times with 1mL buffer C. For experiments with GTP or GDP, buffer C was supplemented with 5mM GTP/GDP. Proteins were eluted with 100μL elution buffer (100 mM Tris pH 8.0, 150mM NaCl, 1mM EDTA, 50mM biotin). Samples were prepared in SDS-PAGE loading buffer (60mM Tris pH 6.8, 2% SDS, 10% glycerol, 0.005% bromophenol blue, 5 mM EDTA) with or without 100mM DTT (final concentration) as indicated. In all SDS–PAGE experiments, equivalent volumes of loading and wash fractions and two-fold more of the elution fraction were loaded and gels stained with Coomassie Brilliant Blue and subsequently analyzed by immunoblotting.

## AlphaFold structural models

AlphaFold and AlphaFold-multimer structure prediction was done with the ColabFold pipeline [41–43]. ColabFold was executed with default settings where multiple sequence alignments were generated with MMseqs2 [65] and HHsearch [66]. The ColabFold pipeline generates five model ranks. Predicted Local Distance Difference Test (pLDDT) and alignment error (pAE) graphs were generated for each rank with custom Matlab script. Ranking of the models was performed based on combined pLDDT and pAE values, with the best ranked models used for further analysis and presentation. Per residue model accuracy was estimated based on pLDDT values (>90, high accuracy; 70–90, generally good accuracy; 50–70, low accuracy; <50, should not be interpreted) [41]. Relative domain positions were validated by pAE. The pAE graphs indicate the expected position error at residue X if the predicted and true structures were aligned on residue Y; the lower the pAE value, the higher the accuracy of the relative position of residue pairs and, consequently, the relative position of domains/subunits/proteins [41]. Structural alignments and images were generated in Pymol (The PyMOL Molecular Graphics System, Version 1.2r3pre, Schrödinger, LLC). For all models, sequences of full-length proteins were used.

## Bioinformatics

Sequence alignments were done using MUSCLE [67] with default parameters in MEGA7 [68] and alignments were visualized with GeneDoc [69]. Protein domains were identified using SMART [70]. % similarity/identity between protein homologs were calculated using EMBOSS Needle software (pairwise sequence alignment) [71].

## Statistics

Statistics were performed using a two-tailed Student's *t*-test for samples with unequal variances.

## Supporting information

**S1 Fig. The RomY protein and the *romY* locus. A.** Sequence alignment of RomY homologs. In red, the conserved N-terminal region, and light blue, the partially conserved C-terminal motif. **B.** The *romY* locus is partially conserved. Transcription direction is indicated by the orientation of arrows with MXAN numbers indicated for the *romY* locus in *M. xanthus*. Note that in the NCBI Reference Sequence NC_008095.1, MXAN_5746 to MXAN_5752 are reannotated as MXAN_RS27850 to MXAN_RS27880; % similarity/identity between homologs from *M. xanthus* and other species is indicated by numbers in the arrows. For the proteins encoded by genes flanking *romY* in *M. xanthus*, domains were identified using SMART [70]. % similarity/identity between protein homologs were calculated using EMBOSS Needle software (pairwise sequence alignment). All listed species belong to the order Myxococcales except for *Bdellovibrio bacteriovorus* HD100 that belongs to the class Oligoflexia and *Flexistipes sinusarabica* and *Calditerrivibrio nitroreducens* that belong to the class Deferribacteres.
(TIF)

**S2 Fig. Lack of RomY causes hyper-reversals independently of the Frz system.** Cells were incubated on 1.5% agar supplemented with 0.5% CTT to score gliding motility. Single data point are plotted in gray. Boxplots of reversals per cell in 15 min; boxes enclose 25th and 75th percentiles, cyan lines indicate the median, and whiskers the 10th and 90th percentiles. In all panels, * $P < 0.01$, two-sided Student's *t*-test.
(EPS)

**S3 Fig. RomY interaction with MglA-GTP and MglB is only detected after DSP cross-linking. A.** SDS-PAGE analysis of purified proteins used in *in vitro* assays. ~5-20ng of the indicated purified proteins were separated by SDS-PAGE and gels stained with Coomassie Brilliant Blue. Calculated molecular weight of the different proteins is indicated. Molecular size markers are indicated on the left. **B.** RomY interaction with MglA-GTP and MglB is not detected in the absence of DSP cross-linking. Proteins were mixed with final concentrations and 10mM GTP/GDP as indicated in the schematics for 30min at RT, and proteins applied to Strep-Tactin coated magnetic beads. Fractions before loading (L), the last wash (W) and after elution (E) were separated by SDS–PAGE, gels stained with Coomassie Brilliant Blue (upper panels) and subsequently probed with α-His$_6$ antibodies (lower panels). All samples were prepared with loading dye supplemented with 100mM DTT. For each combination, fractions were separated on the same gel. Gaps between lanes indicate lanes deleted for presentation purposes. **C.** RomY interacts with MglA-GTP and MglB. Proteins were mixed with final concentrations and 10mM GTP/GDP as indicated in the schematics for 30min at RT, DSP added (final concentration 200μM, 5min, RT), DSP quenched, and proteins applied to Strep-Tactin coated magnetic beads. Fractions before loading (L), the last wash (W) and after elution (E) were separated by SDS–PAGE, gels stained with Coomassie Brilliant Blue (upper panels) and subsequently probed with α-MglA or α-MglB antibodies (lower panels). Eluted samples were treated with loading buffer with (+) or without (-) 100 mM DTT to break protein cross-links as indicated under the gels and immunoblots. For each combination, fractions were separated on the same gel. Gaps between lanes indicate lanes deleted for presentation purposes. Note that the experiments in Fig 2C are similar to those presented in this panel in which proteins were separated on a 7.5% acrylamide gel; proteins were separated on an 8–16% acrylamide

gradient gel in the experiments in Fig 2C to increase the separation of lower molecular weight proteins. **D.** RomY does not interact with His$_6$-MalE after DSP cross-linking. Proteins were mixed as in C. Fractions before loading (L), the last wash (W) and after elution (E) were separated by SDS–PAGE, gels stained with Coomassie Brilliant Blue (upper panels) and subsequently probed with α-His$_6$ antibodies (lower panels). All samples were treated with loading buffer containing 100mM DTT to break cross-links before SDS-PAGE. For each combination, fractions were separated on the same gel. Gaps between lanes indicate lanes deleted for presentation purposes. Proteins were separated on an 8–16% acrylamide gradient gel.
(TIF)

**S4 Fig. AlphaFold and AlpfaFold-Multimer models of RomY, MglA:RomY, (MglB)$_2$: RomY, MglA:(MglB)$_2$:RomY and MglA:(MglB)$_2$. A.** pLDDT and pAE plots of five models of the indicated protein and complexes. The model rank shown in Fig **3A, 3B** and **3B–3E** are indicated by a green box. AlphaFold was used for modeling RomY and AlphaFold-Multimer for modeling the complexes. **B.** AlphaFold-Multimer model of MglA:(MglB)$_2$ superimposed on the solved structure of MglAGTPγS:(MglB)$_2$ (pdb ID code: 6izw [30]). The AlphaFold model is in light green and the solved structures of MglAGTPγS and of (MglB)$_2$ in cyan and red, respectively. The green sphere indicates Mg$^{2+}$. AlphaFold model rank 1 is shown. **C.** AlphaFold-Multimer model of MglA:RomY. Left panel, MglA is in yellow, the N-terminal domain of RomY in light teal, and the remainder of RomY in cyan. Middle panel, as in left panel except that only the N-terminal domain of RomY is shown. Right panel, superimposition of MglA from the model of the MglA:RomY complex with the solved structure of MglAGTPγS in light green (pdb ID: 6h17 [32]). In the modeled structure of MglA, the P-loop is in purple, switch-1 in blue and switch-2 in green. AlphaFold model rank 3 is shown. **D.** AlphaFold-Multimer model of (MglB)$_2$:RomY. Right panel, the MglB homodimer is in red and the N-terminal domain of RomY in light teal, and the remainder of RomY in cyan. Middle panel, as in left panel except that only the N-terminal domain of RomY is shown. Right panel, superimposition of the MglB homodimer from the model of the (MglB)$_2$:RomY complex with the solved structure of the MglB homodimer in light green (pdb ID code: 6hjm [32]). AlphaFold model rank 1 is shown. **E.** Alphafold-Multimer model of MglA:(MglB)$_2$:RomY complex. Left panel, the proteins are colored as in **C-D**. Right panel, superimposition of MglA:(MglB)$_2$ from the MglA:(MglB)$_2$:RomY complex with the solved structure of MglAGTPγS:(MglB)$_2$ (pdb ID code: 6izw [30]). The solved structure is in light green and the AlphaFold model in yellow and red. AlphaFold model rank 3 is shown.
(EPS)

**S5 Fig. The N-terminal domain of RomY has partial RomY activity. A.** The N-terminal domain of RomY has partial RomY activity. Cells were incubated on 0.5/1.5% agar with 0.5% CTT to score T4P-dependent/gliding motility. Scale bars, 1mm (left), 500 μm (middle), 50 μm (right). Data are shown from a representative experiment. **B.** Accumulation of RomY variants. Immunoblot analysis of RomY accumulation. Cell lysates were prepared from same number of cells, separated by SDS–PAGE and probed with α-RomY antibodies and α-PilC antibodies after stripping (loading control). In the two rightmost lanes, 6.0ng and 3.3ng, respectively of purified Strep-RomY and Strep-RomY$^N$ was loaded. RomY$^N$ and Strep-RomY$^N$ have calculated molecular masses of 10.1 and 13.1kDa, respectively.
(EPS)

**S6 Fig. Overproduction of MglB or RomY does not affect speed of cells moving by T4P-dependent motility.** Cells were treated as in Fig 4A and then T4P-dependent single cell motility analyzed. Strains are numbered as in Fig 4A. In the legend, + indicates presence of WT

gene, Δ in-frame deletion, 0/500 μM vanillate concentration, and * the WT grown in the presence of 500μM vanillate. Individual data points from a representative experiment with $n = 20$ cells are plotted in gray. Because the experiment relies on induction of gene expression, protein levels vary slightly between experiments, making the direct comparison between biological replicates difficult. Consequently, data from only one representative experiment is shown. The cells analyzed are the same as in Fig 4**B**. Boxplots are as in Fig 1**D**.
(EPS)

**S7 Fig. Analysis of RomY-YFP and MglB-mVenus fusions and accumulation levels of RomY-YFP, MglA-mVenus, SgmX-mVenus AglZ-YFP and MglB-mVenus. A**. RomY-YFP and MglB-mVenus are functional fusions. Motility assays were done as in Fig 1**B**. Scale bars, 1mm (0.5% agar) and 50μm (1.5% agar). **B**. RomY-YFP accumulation. Immunoblot analysis was done as in Fig 1**C**. RomY and RomY-YFP with calculated molecular masses are indicated. PilC served as a loading control. **C-F**. MglA-mVenus, SgmX-mVenus, AglZ-YFP and MglB-mVenus accumulation. Immunoblot analysis was done as in Fig 1**C**. Relevant proteins with their with calculated molecular masses are indicated. PilC served as a loading control.
(EPS)

**S1 Table. Genomic distribution of *mglA*, *mglB*, *romR*, *romX* and *romY* in a set of 1611 prokaryotic genomes.**
(XLSX)

**S2 Table. *M. xanthus* strains used in this work.**
(DOCX)

**S3 Table. Plasmids used in this work.**
(DOCX)

**S4 Table. Primers used in this work.**
(XLSX)

## Acknowledgments

We gratefully acknowledge the help of Marco Herfurth and Dr. Dorota Skotnicka for many helpful discussions and of Dr. Kristin Wuichet and Dr. Daniela Keilberg in the identification and initial characterization of RomY.

## Author Contributions

**Conceptualization:** Dobromir Szadkowski, Luís António Menezes Carreira, Lotte Søgaard-Andersen.

**Data curation:** Dobromir Szadkowski.

**Formal analysis:** Dobromir Szadkowski, Luís António Menezes Carreira.

**Funding acquisition:** Lotte Søgaard-Andersen.

**Investigation:** Dobromir Szadkowski, Luís António Menezes Carreira.

**Methodology:** Dobromir Szadkowski, Luís António Menezes Carreira.

**Project administration:** Lotte Søgaard-Andersen.

**Resources:** Lotte Søgaard-Andersen.

**Software:** Dobromir Szadkowski.

**Supervision:** Lotte Søgaard-Andersen.

**Validation:** Dobromir Szadkowski, Luís António Menezes Carreira.

**Visualization:** Dobromir Szadkowski.

**Writing – original draft:** Dobromir Szadkowski, Lotte Søgaard-Andersen.

**Writing – review & editing:** Dobromir Szadkowski, Luís António Menezes Carreira, Lotte Søgaard-Andersen.

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
