## [Decision Letter · Decision Letter 0]

19 May 2022

Dear Dr Søgaard-Andersen,

Thank you very much for submitting your Research Article entitled 'A bipartite, low-affinity roadblock domain-containing GAP complex regulates bacterial front-rear polarity' to PLOS Genetics.

The manuscript was fully evaluated at the editorial level and by independent peer reviewers. The reviewers appreciated the attention to an important problem, but raised some substantial concerns about the current manuscript. I agree with Reviewer #2 that the claim that RomY out competes RomRX is not fully supported by the data presented. In addition, the interaction between MglB and RomY needs to be experimentally validated. Based on the reviews, we will not be able to accept this version of the manuscript, but we would be willing to review a much-revised version. We cannot, of course, promise publication at that time.

If you decide to revise the manuscript for further consideration at PLOS Genetics, please aim to resubmit within the next 60 days, unless it will take extra time to address the concerns of the reviewers, in which case we would appreciate an expected resubmission date by email to plosgenetics@plos.org.

[LINK]

We are sorry that we cannot be more positive about your manuscript at this stage. Please do not hesitate to contact us if you have any concerns or questions.

Yours sincerely,

Beiyan Nan, Ph.D.

Guest Editor

PLOS Genetics

Gregory P. Copenhaver

Editor-in-Chief

PLOS Genetics

Reviewer's Responses to Questions

**Comments to the Authors:**

Reviewer #1: The determinants of front-rear polarity in M. xanthus motility have been extensively studied over the past years. However it remained unknown how the GAP or GEF activity on MglA (the output of the polarity module) is dominant at the lagging or leading cell pole, respectively, while both the GEF and the GAP proteins (or protein complex in the case of the GEF) displayed the same asymmetric distribution at both poles. The authors investigated how the GEF and GAP activities are spatially separated and describe the protein RomY as weak interaction partner of MglB and a stimulator of its GAP function (“co-GAP”). The asymmetric localisation of RomY supports its co-GAP role being restricted at the lagging cell pole. While MglB is important for the recruitment of RomY, additional, unknown determinants of RomY localisation still remain to be discovered.

Overall this is a carefully conducted study, which combines bacterial genetics, microscopy, motility assays, enzymatic assays, revealing a new piece in the puzzle of Myxococcus polarity factors. Future research will be needed to decipher how the co-GAP function of RomY is achieved. The manuscript is well written and easy to read. I only have minor comments.

Figure 1B: only one scale bar is shown while the legend indicates scale bars for left, middle and right.

Figure 2A: did the authors test whether the Strep-tagged RomY (used in in vitro assay to conclude on its GAP stimulating activity) complements the ∆romY motility defects similarly as the untagged version?

Figure 2B: to support the interaction, it would have been nice to migrate side-by-side the crosslinked samples before and after reduction with DTT. The authors only show the post-DTT samples, while pre-DTT migration would have allowed to detect a band at higher MW corresponding to the expected size of the complex.

Figure 5C: providing the complementation experiment would support the claim (role of RomY on MglA polar localisation)

Figure 5D: it is unclear which strain/genotype is used here.

At several places (e.g. lines 425, 426), the authors claim that MglB GAP activity is only stimulated at the lagging pole. Even though RomY is more prominently localised at the lagging pole, it can’t be excluded that the little fraction present at the leading pole could still turn on MglB GAP activity there as well (though to a lower extend, according to the weak interaction data). I would rather think that there is an overall dominance of the GEF activity over the GAP activity on the population of MglA proteins at the leading pole (and the opposite at the lagging pole), more than an exclusive GAP or GEF activity at either pole.

Text:

- Rephrase title of S7 Fig.

- Line 454: adds instead of add.

- S3 Fig. Legend: cross instead of crosss; same at line 977

Reviewer #2: The manuscript by Szadkowski et. al. describes the identification of a new component, RomY, in the front-rear polarity establishment and reversals in Myxococcus xanthus. They term RomY as a ‘co-GAP’ that activates the GAP action by MglB for stimulating the GTPase activity by MglA further. The authors present detailed in-vivo and some preliminary in-vitro experiments to validate their model of polarity regulation. Some of the claims regarding competition and low affinities of complex formation would require a finer set of experiments involving GTPase measurements at various stoichiometries of the interacting partners. The claim that RomRX GEF activity is outcompeted, again, remains a speculation unless proved by competition assays involving RomRX vs RomY.

Please see below some major and minor clarifications, to be sorted for improvement of the manuscript.

Major points:

1. For RomY to function as a coGAP, it appears that there is no stoichiometric requirement, based on the claim that overexpression of only one of the proteins compared to the other affected the activities, as observed in vivo. How can this be explained mechanistically?

2. Please provide a discussion or a summary table in the supplementary to comment on how these GTPase measurements for MglA compare with the existing published literature that reported the GTPase activities.

3. It is mentioned that RomY outcompetes RomRX activity. The possibilities of a competition between these two proteins upon interaction with MglAB has not been discussed. The proposed mechanism does not explain how RomY acts as a competitor for RomRX. Hence, the question of restriction of the GEF activity remains unanswered.

4. Validation of the AlphaFold model should be discussed.

- One of the possibilities is to use the crosslinking data (used for capturing the transient complexes in pull-down assays) and identify the crosslinked peptides. This also will demonstrate that the captured interactions are indeed specific.

- The AlphaFold model shows that RomY-MglB interface consists mainly the C-terminal helix of MglB. Will a GTPase assay with the deletion construct of C-terminal helix of MglB help in confirming the mechanism of action of the RomY co-GAP and validate the structure proposed?

5. How would you define the concept of a co-GAP? Given that the active site residues are present in MglA itself, and MglB is capable of orienting these, how do you propose that the co-GAP contributes to the increased activity?

6. How do the GTPase activity levels vary with varying ratios of RomY and MglB? As of now, a 1:2 ratio of MglB has been used, and a 1:2 ratio of RomY too. The stoichiometry at which the GTPase activities are saturated will be interesting to see, in order to get estimates of stoichiometry.

7. Has the oligomeric status of RomY been checked? Why was a monomeric state assumed for generating the AlphaFold model? Will alternative models be generated in AlphaFold if RomY was considered as a dimer?

Specific questions/clarifications:

1. Are there any estimates of concentration of MglB at the poles, and within the cell? How do the differences in concentration at the two poles correlate with the affinity explanation?

2. MXAN5749 in UNIProt has been given as 258 aa protein. Explain the discrepancy between the reported sequence in the paper and the database. Since there is a Western blot analysis of the native protein, please provide the protein ladder information so that the size is confirmed.

3. Line 192-193: If RomY inhibits RomR/RomX GEF activity, then ΔromY mutation would not suppress the gliding defect in the ΔromR and ΔromX mutants, So it can act as an inhibitor of RomRX GEF, and this inhibition can happen in an MglB dependent manner. So it is not really clear why scenario 1 is excluded.

4. Line 211: Why was this assay not done in a 1:2:1 ratio of MglA:MglB:RomY, given that your model proposes a 1:2:1 complex? What happens to the GTPase rate in such a case? Again, is there a further increase in GTPase activity in vitro with increasing concentrations of MglB, beyond 1:2? This experiment is essential to get information on the maximum activity that can be achieved, when MglB is in saturating levels. Similarly, how does concentration variation of RomY affect the activity?

5. Was a pull-down attempted with MglA-GTP-MglB complex without or with the crosslinker? According to the proposed model, an interaction with the MglAB complex might be a more stable interaction, and might be retained even in the absence of a crosslinker.

6. Line 204 mentions His6-MglB while 206 mentions MglB-His6. What is the position of the tag? Please correct the text accordingly.

7. RomY could increase the affinity of MglB homo-dimer for MglA. This statement implies that an increased concentration of MglB should result in an increase of GTPase activity independent of RomY addition. The GTPase activity measurements with increased MglB concentrations (increasing ratios till it reaches saturation) will help in coming up with a mechanism of co-GAP. An explanation of how this does not happen within the cell should be provided.

8. Line 278 - 280: MglA and MglB form a tight complex, which can be retained in SEC experiments and with KD of about 1 uM. This also results in a stimulation of MglA GTPase activity, and efficiently orients the catalytic residues. Mechanistically, the requirements of a coGAP is not clear.

MglA active site is well formed with just the interaction with MglB. Hence, the role of RomY could be to further stabilise the MglA/MglB interaction? Since there is no requirement for a supply of active site residue for the GTPase from the GAP or co-GAP, the function of the co-GAP should be stabilization of the active site conformation, a role proven to be performed by MglB alone.

In this context, how can the requirement of RomY be explained? Having a approximate estimates of protein concentrations at the poles during reversals will be interesting information, if available.

9. Line 309: It is not entirely clear why an overproduction of MglB in the absence of RomY will not affect reversal frequencies? We do observe a dose dependent activation of the MglA GTPase activity by MglB (Zhang et al 2010, Fig 1C). Kindly clarify why it doesn't have an effect in-vivo. Considering the mechanism of activation by a co-GAP based on stabilisation of interaction between MglA and MglB, it seems surprising that this effect is not observed.

10. Line 327: “In the absence of RomR, RomY-YFP polar localization was also decreased”. Elaborate on the role of RomR on RomY localisation. Why do you observe such a phenotype? This has not been discussed further.

Line 327: Is there a decrease in RomY levels in the absence of RomR (the band in Figure S7 looks less compared to the others). Is this significant over multiple repeats? Can this be a cause for the decrease in polar localization?

11. RomY native levels with overexpression of MglB leads to low MglA-GTP levels: How will this compensate for the increased requirement of 1 RomY per MglAB complex formed (according to the proposed AlphaFold model)? Please clarify this in the text.

12. Line 343: “in the absence of RomY, polar localization of MglA-mVenus was increased and switched toward more bipolar symmetric”. Does this observation actually provide proof for showing that inhibition of RomRX at the lagging pole was lifted hence RomRX can recruit MglA-GTP to both the poles, validating Scenario-1 in line 193?

13. Lines 359-361: Most of the data points for MglB-mVenus seems to be at a unipolar localization? Why is it given that MglB-mVenus is bipolar asymmetric?

14. Line 394: Even when MglB was overproduced, its activity was dependent on RomY. This does not corroborate with just a requirement for stabilization of the complex. Given the stoichiometry proposed in the model, a 20-fold overproduction of MglB should in principle help in attaining the highest turnover number for MglA, and compensate for any decrease in GTPase activity due to dissociation of the MglA/MglB complex.

15. Line 404 - 406: ‘formation of the complex rather than protein concentrations’ is an ambiguous statement, any complex formation, irrespective of the affinities is dependent on protein concentrations. I wonder if the authors imply stoichiometries here? This sentence has been included in various instances in the manuscript and might need to be rephrased.

16. If the MglA-GTP-RomY interaction occurs as proposed in the AlphaFold model, RomY should be able to stabilise the binding of GTP to MglA. Has this been tested experimentally?

17. Line 417: “this localization pattern is ideal to allow stable and switchable polarity”. It is still not clear why the GEF complex would accumulate majorly at the lagging pole only to be outcompeted by the GAP complex? This is very unlike what is usually seen as per the localised activity of the eukaryotic GEF-GTPase.

Minor points:

1. Fig 1E. 4th lane (delMglA) has not been marked as NA.

2. Line 91: MglA GDP is diffused in the cytoplasm

3. Line 182: How can it be concluded that RomY acts downstream of the Frz system? Though I understand that it is related to MglAB and RomXY, without epistasis data involving Frz genes, the claim in this sentence cannot be made. This should be mentioned as a hypothesis.

4. Lines 241-245: AlphaFold predictions for existing structures are usually very reliable because the structural information is used in generating the prediction. In general, AlphaFold also tends to pick up conformations which are more represented in the PDB. Hence, it is not accurate to claim that RomY interacts with the GTP-bound conformation based on the AlphaFold model.

5. Line 267: “RomYN stimulated MglA GTPase activity in the presence of MglB almost as efficiently as full-length RomY”. However, it appears from Fig 2A, that the difference between the activities of RomYN and RomY full length might be significant?

6. Line 301: “We conclude that MglB and RomY accumulate independently of each other “ This might be confused with MglB dependent localisation of RomY, and hence should be rephrased to remove ambiguity.

**Have all data underlying the figures and results presented in the manuscript been provided?**

Reviewer #1: None

Reviewer #2: Yes

PLOS authors have the option to publish the peer review history of their article (what does this mean?). If published, this will include your full peer review and any attached files.

Reviewer #1: No

Reviewer #2: No

---

## [Decision Letter · Decision Letter 1]

17 Aug 2022

Dear Dr Søgaard-Andersen,

We are pleased to inform you that your manuscript entitled "A bipartite, low-affinity roadblock domain-containing GAP complex regulates bacterial front-rear polarity" has been editorially accepted for publication in PLOS Genetics. Congratulations!

Yours sincerely,

Beiyan Nan, Ph.D.

Guest Editor

PLOS Genetics

Gregory P. Copenhaver

Editor-in-Chief

PLOS Genetics

Comments from the reviewers (if applicable):

Reviewer's Responses to Questions

**Comments to the Authors:**

Reviewer #1: I would like to thank the authors for addressing my comments and improving their manuscript accordingly by textual clarifications and additional experiments.

Reviewer #2: I am satisfied with the changes made in the revision and recommend publication without any further changes.

**Have all data underlying the figures and results presented in the manuscript been provided?**

Reviewer #1: None

Reviewer #2: Yes

PLOS authors have the option to publish the peer review history of their article (what does this mean?). If published, this will include your full peer review and any attached files.

Reviewer #1: No

Reviewer #2: No

**Data Deposition**

http://datadryad.org/submit?journalID=pgenetics&manu=PGENETICS-D-22-00441R1

**Press Queries**

---

## [Editor Report · Acceptance letter]

30 Aug 2022

PGENETICS-D-22-00441R1 

A bipartite, low-affinity roadblock domain-containing GAP complex regulates bacterial front-rear polarity 

Dear Dr Søgaard-Andersen, 

We are pleased to inform you that your manuscript entitled "A bipartite, low-affinity roadblock domain-containing GAP complex regulates bacterial front-rear polarity" has been formally accepted for publication in PLOS Genetics! Your manuscript is now with our production department and you will be notified of the publication date in due course.

With kind regards,

Zsofia Freund

PLOS Genetics

On behalf of:
